# Intergenerational adaptations to stress are evolutionarily conserved, stress-specific, and have deleterious trade-offs

Nicholas O Burton[1,2,3]*, Alexandra Willis[4], Kinsey Fisher[5], Fabian Braukmann[2], Jonathan Price[2], Lewis Stevens[6,7], L Ryan Baugh[5,8], Aaron Reinke[4], Eric A Miska[2,7,9]

[1]Centre for Trophoblast Research, Department of Physiology, Development and Neuroscience, University of Cambridge, Cambridge, United Kingdom; [2]Gurdon Institute, University of Cambridge, Cambridge, United Kingdom; [3]Van Andel Institute, Grand Rapids, United States; [4]Department of Molecular Genetics, University of Toronto, Toronto, Canada; [5]Department of Biology, Duke University, Durham, United States; [6]Department of Molecular Biosciences, Northwestern University, Evanston, United States; [7]Wellcome Sanger Institute, Wellcome Genome Campus, Cambridge, United Kingdom; [8]Center for Genomic and Computational Biology, Duke University, Durham, United States; [9]Department of Genetics, University of Cambridge, Cambridge, United Kingdom

*For correspondence:
nick.burton@vai.org

Competing interest: The authors declare that no competing interests exist.

**Abstract** Despite reports of parental exposure to stress promoting physiological adaptations in progeny in diverse organisms, there remains considerable debate over the significance and evolutionary conservation of such multigenerational effects. Here, we investigate four independent models of intergenerational adaptations to stress in *Caenorhabditis elegans* – bacterial infection, eukaryotic infection, osmotic stress, and nutrient stress – across multiple species. We found that all four intergenerational physiological adaptations are conserved in at least one other species, that they are stress -specific, and that they have deleterious tradeoffs in mismatched environments. By profiling the effects of parental bacterial infection and osmotic stress exposure on progeny gene expression across species, we established a core set of 587 genes that exhibited a greater than twofold intergenerational change in expression in response to stress in *C. elegans* and at least one other species, as well as a set of 37 highly conserved genes that exhibited a greater than twofold intergenerational change in expression in all four species tested. Furthermore, we provide evidence suggesting that presumed adaptive and deleterious intergenerational effects are molecularly related at the gene expression level. Lastly, we found that none of the effects we detected of these stresses on *C. elegans* F1 progeny gene expression persisted transgenerationally three generations after stress exposure. We conclude that intergenerational responses to stress play a substantial and evolutionarily conserved role in regulating animal physiology and that the vast majority of the effects of parental stress on progeny gene expression are reversible and not maintained transgenerationally.

## Introduction

Multigenerational effects of a parent's environment on progeny have been reported to contribute to numerous organismal phenotypes and pathologies in species ranging from plants to mammals (*Agrawal et al., 1999*; *Bozler et al., 2019*; *Burton et al., 2020*; *Burton et al., 2017*; *Dantzer et al., 2013*; *Dias and Ressler, 2014*; *Hibshman et al., 2016*; *Houri-Zeevi et al., 2020*; *Jordan et al., 2019*; *Kaletsky et al., 2020*; *Kishimoto et al., 2017*; *Klosin et al., 2017*; *Luna et al., 2012*; *Ma et al., 2019*; *Moore et al., 2019*; *Öst et al., 2014*; *Palominos et al., 2017*; *Posner et al.,*

*2019*; *Veenendaal et al., 2013*; *Vellichirammal et al., 2017*; *Webster et al., 2018*; *Wibowo et al., 2016*; *Willis et al., 2021*). These effects on progeny include many notable observations of intergenerational (lasting 1–2 generations) adaptive changes in phenotypically plastic traits such as the development of wings in pea aphids (*Vellichirammal et al., 2017*), helmet formation in *Daphnia* (*Agrawal et al., 1999*), accelerated growth rate in red squirrels (*Dantzer et al., 2013*), and physiological adaptations to osmotic stress and pathogen infection in both *Arabidopsis* (*Luna et al., 2012*; *Wibowo et al., 2016*) and *Caenorhabditis elegans* (*Burton et al., 2020*; *Burton et al., 2017*). These intergenerational adaptive changes in development and physiology, which include effects that are sometimes interchangeably described as parental effects, can lead to substantial increases in organismal survival, with up to 50-fold increases in the survival of offspring from stressed parents being reported when compared to the offspring from naive parents (*Burton et al., 2020*). While many of the most studied intergenerational effects of a parent's environment on offspring have been identified in plants and invertebrates, intergenerational effects have also been reported in mammals (*Dantzer et al., 2013*; *Dias and Ressler, 2014*). Similar to findings in plants and invertebrates, some observations of intergenerational effects in mammals have been found to be physiologically adaptive (*Dantzer et al., 2013*), but many others, such as observations of fetal programming in humans (*de Gusmão Correia et al., 2012*; *Langley-Evans, 2006*; *Schulz, 2010*) and studies of the Dutch Hunger Winter (*Veenendaal et al., 2013*), have been reported to be deleterious. Nonetheless, even for these presumed deleterious intergenerational effects, it has been hypothesized that under different conditions the intergenerational effects of fetal programming, such as the effects caused by the Dutch Hunger Winter, might be considered physiologically adaptive (*Hales and Barker, 2001*; *Hales and Barker, 1992*).

If intergenerational responses to environmental stresses represent evolutionarily conserved processes, if they are general or stress-specific effects, and whether adaptive and deleterious intergenerational effects are molecularly related remains unknown. Furthermore, multiple different studies have recently reported that some environmental stresses elicit changes in progeny physiology and gene expression that persist for three or more generations, also known as transgenerational effects (*Kaletsky et al., 2020*; *Klosin et al., 2017*; *Ma et al., 2019*; *Moore et al., 2019*; *Posner et al., 2019*; *Webster et al., 2018*). However, if intergenerational effects (lasting 1–2 generations) and transgenerational effects (lasting 3+ generations) represent related or largely separable phenomena remains unclear. Answering these questions is critically important not only in understanding the role that multigenerational effects play in evolution, but also in understanding how such effects might contribute to multiple human pathologies that have been linked to the effects of a parent's environment on offspring, such as Type 2 diabetes and cardiovascular disease (*Langley-Evans, 2006*).

Here, we investigated the evolutionary conservation, stress specificity, and potential tradeoffs of four independent models of intergenerational adaptations to stress in *C. elegans* – bacterial infection, eukaryotic infection, nutrient stress, and osmotic stress. We found that all four models of intergenerational adaptive effects are conserved in at least one other species, but that all exhibited a different pattern of evolutionary conservation. Each intergenerational adaptive effect was stress-specific and multiple intergenerational adaptive effects exhibited deleterious tradeoffs in mismatched environments or environments where multiple stresses were present simultaneously. By profiling the effects of multiple different stresses on offspring gene expression across species we identified a set of 37 genes that exhibited intergenerational changes in gene expression in response to stress in all species tested. In addition, we found that an inversion in the expression of a key gene involved in the intergenerational response to bacterial infection, *rhy-1*, from increased expression to decreased expression in the offspring of stressed parents, correlates with an inversion of an adaptive intergenerational response to bacterial infection in *C. elegans* and *C. kamaaina* to a deleterious intergenerational effect in *C. briggsae*. Lastly, we report that none of the effects of multiple different stresses on F1 gene expression that we detected here persisted transgenerationally into F3 progeny in *C. elegans*. Our findings demonstrate that intergenerational adaptive responses to stress are evolutionarily conserved, stress-specific, and are predominantly not maintained transgenerationally. In addition, our findings suggest that the mechanisms that mediate intergenerational adaptive responses in some species might be related to the mechanisms that mediate intergenerational deleterious effects in other species.

## Results

### Intergenerational adaptations to stress are evolutionarily conserved

To test if any of the intergenerational adaptations to stress that have been reported in *C. elegans* are evolutionarily conserved in other species we focused on four recently described intergenerational adaptations to abiotic and biotic stresses – osmotic stress (*Burton et al., 2017*), nutrient stress (*Hibshman et al., 2016*; *Jordan et al., 2019*), *Pseudomonas vranonvensis* infection (bacterial) (*Burton et al., 2020*), and *Nematocida parisii* infection (eukaryotic – microsporidia) (*Willis et al., 2021*). All of these stresses are exclusively intergenerational and did not persist beyond two generations in any experimental setup previously analyzed (*Burton et al., 2017*; *Burton et al., 2020*; *Willis et al., 2021*). We tested if these four intergenerational adaptive responses were conserved in four different species of *Caenorhabditis* (*C. briggsae*, *C. elegans*, *C. kamaaina*, and *C. tropicalis*) that shared a last common ancestor approximately 30 million years ago and have diverged to the point of having approximately 0.05 substitutions per site at the nucleotide level (*Figure 1A*; *Cutter, 2008*). These species were chosen because they represent multiple independent branches of the *Elegans* group (*Figure 1A*) and because we could probe the conservation of underlying mechanisms using established genetics approaches.

We exposed parents of all four species to *P. vranovensis* and subsequently studied their offspring's survival rate in response to future *P. vranovensis* exposure. We found that parental exposure to the bacterial pathogen *P. vranovensis* protected offspring from future infection in both *C. elegans* and *C. kamaaina* (*Figure 1B*) and that this adaptive intergenerational effect in *C. kamaaina* required the same stress response genes (*cysl-1* and *rhy-1*) as previously reported for *C. elegans* (*Burton et al., 2020*; *Figure 1C*), indicating that these animals intergenerationally adapt to infection via a similar and potentially conserved mechanism. By contrast, we found that naive *C. briggsae* animals were more resistant to *P. vranovensis* than any of the other species tested, but exposure of *C. briggsae* parents to *P. vranovensis* caused greater than 99 % of offspring to die upon future exposure to *P. vranovensis* (*Figure 1B*). We confirmed that parental *P. vranovensis* exposure resulted in an adaptive intergenerational effect for *C. elegans* but a deleterious intergenerational effect for *C. briggsae* by testing multiple additional wild isolates of both species (*Figure 1—figure supplement 1A-C*). Parental exposure to *P. vranovensis* had no observable effect on offspring response to infection in *C. tropicalis* (*Figure 1B*). We conclude that parental exposure to *P. vranovensis* causes substantial changes in offspring susceptibility to future *P. vranovensis* exposure in multiple species, but whether those effects are protective or deleterious for offspring is species-dependent.

Using a similar approach to investigate intergenerational adaptive responses to other stresses, we found that parental exposure to mild osmotic stress protected offspring from future osmotic stress in *C. elegans*, *C. briggsae*, and *C. kamaaina*, but again not in *C. tropicalis* (*Figure 1D*). This intergenerational adaptation to osmotic stress in *C. briggsae* and *C. kamaaina* required the glycerol-3-phosphate dehydrogenase *gpdh-2* (*Figure 1E* and *Figure 1—figure supplement 1D*), similar to previous observations for *C. elegans* (*Burton et al., 2017*) and indicating that these adaptations are regulated by similar and likely evolutionarily conserved mechanisms.

We then sought to test if intergenerational resistance to infection by the eukaryotic pathogen *N. parisii* is similarly conserved in *Caenorhabditis* species. *N. parisii* is a common natural pathogen of both *C. elegans* and *C. briggsae* (*Zhang et al., 2016*). Here, we show that *N. parisii* can also infect *C. kamaaina* and *C. tropicalis* (*Figure 1—figure supplement 1E-G*). By investigating the effects of parental *N. parisii* infection on offspring across species, we found that parental exposure of *C. elegans* and *C. briggsae* to *N. parisii* protected offspring from future infection (*Figure 1F*). By contrast, parental exposure of *C. kamaaina* and *C. tropicalis* to *N. parisii* had no observable effect on offspring infection rate (*Figure 1F*).

Lastly, we investigated the intergenerational effects of nutrient stress on offspring. We found that parental nutrient stress by food deprivation resulted in larger offspring in both *C. elegans* and *C. tropicalis*, which is predicted to be adaptive (*Hibshman et al., 2016*), but had minimal effects on offspring size in *C. briggsae* and *C. kamaaina* (*Figure 1G*). Collectively, our findings indicate that all four reported intergenerational adaptive effects in *C. elegans* are conserved in at least one other species but all four show a different pattern of conservation, which is consistent with each response being regulated by distinct molecular mechanisms (*Burton et al., 2020*; *Burton et al., 2017*; *Hibshman et al., 2016*; *Jordan et al., 2019*; *Willis et al., 2021*).

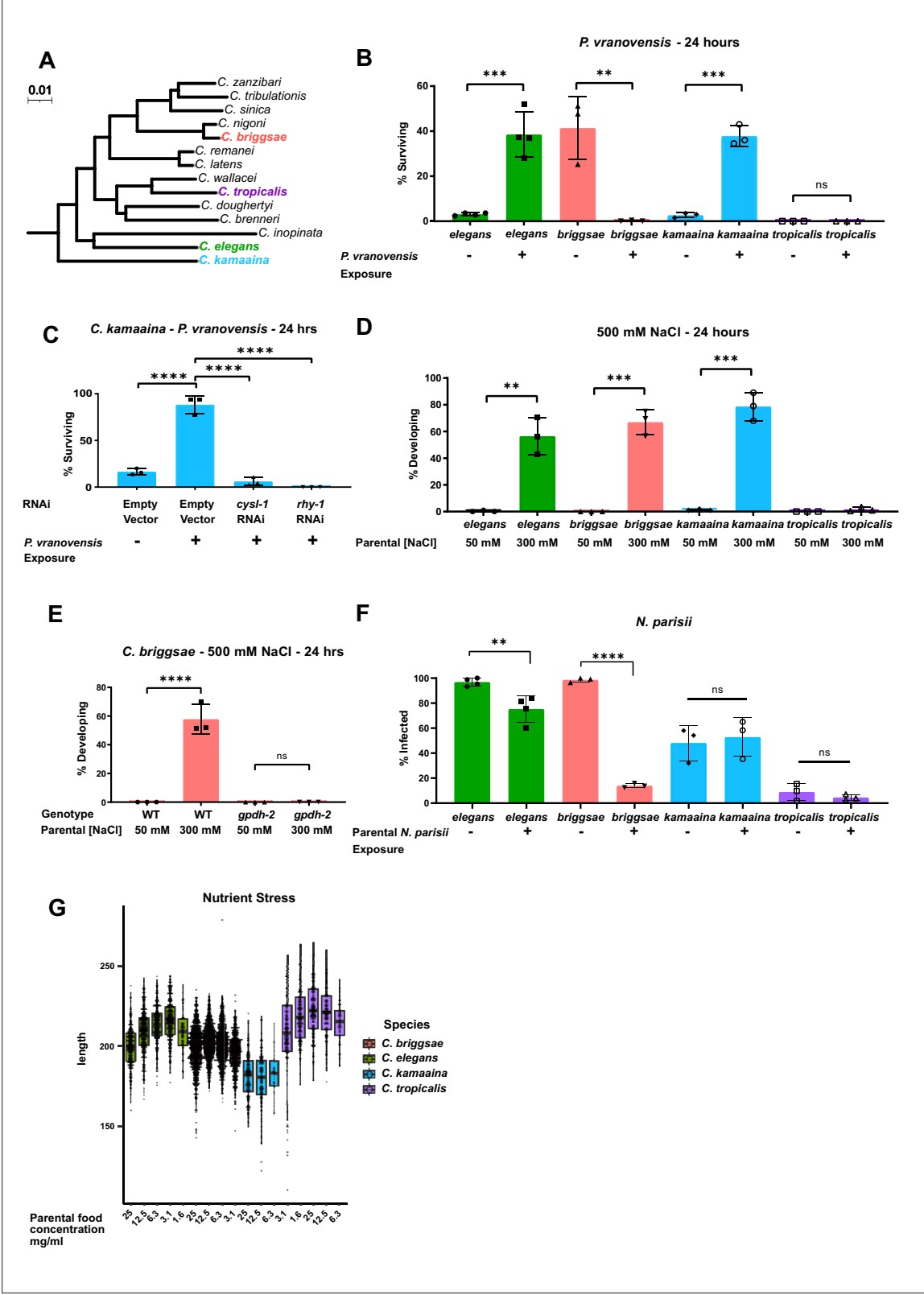

**Figure 1.** Intergenerational adaptations to multiple stresses are evolutionarily conserved in multiple species of *Caenorhabditis*. (**A**) Phylogenetic tree of the *Elegans* group of *Caenorhabditis* species adapted from ***Stevens et al., 2020***. Scale represents substitutions per site. (**B**) Percent of wild-type *C. elegans* (N2), *C. kamaaina* (QG122), *C. briggsae* (AF16), and *C. tropicalis* (JU1373) animals surviving after 24 hr on plates seeded with *P. vranovensis* BIGb0446. Data presented as mean values ± s.d. *n* = 3–4 experiments of >100 animals. (**C**) Percent of *C. kamaaina* wild-type (QG122) animals surviving

*Figure 1 continued on next page*

*Figure 1 continued*

after 24 hr of exposure to *P. vranovensis*. Data presented as mean values ± s.d. *n* = 3 experiments of >100 animals. (**D**) Percent of wild-type animals mobile and developing at 500 mM NaCl after 24 hr. Data presented as mean values ± s.d. *n* = 3 experiments of >100 animals. (**E**) Percent of wild-type and *Cbr-gpdh-2*(*syb2973*) mutant *C. briggsae* (AF16) mobile and developing after 24 hr at 500 mM NaCl. Data presented as mean values ± s.d. *n* = 3 experiments of >100 animals. (**F**) Percent of animals exhibiting detectable infection by *N. parisii* as determined by DY96 staining after 72 hr for *C. elegans* and *C. briggsae*, or 96 hr for *C. kamaaina* and *C. tropicalis*. Data presented as mean values ± s.e.m. *n* = 3–4 experiments of 83–202 animals. (**G**) Boxplots for length of L1 progeny from P0 parents that were subject to the HB101 dose series. Larvae were measured using Wormsizer. Boxplots show median length with four quartiles. *n* = 3–8 experiments of 50–200 animals. **p < 0.01, ***p < 0.0001, ****p < 0.0001.

The online version of this article includes the following figure supplement(s) for figure 1:

**Source data 1.** Statistics source data for *Figure 1*.

**Figure supplement 1.** Intergenerational responses to environmental stress are conserved in wild isolates of *Caenorhabditis* species.

## Parental exposure to environmental stresses leads to common and stress-specific gene expression changes in offspring across species

Of the four intergenerational models investigated here, parental exposure of *C. elegans* to osmotic stress and *P. vranovensis* infection were previously reported to cause substantial changes in offspring gene expression, including the increased expression of genes that are required for the observed intergenerational adaptations (*Burton et al., 2020*; *Burton et al., 2017*). These effects of parental stress exposure on offspring gene expression resemble a subset of the transcriptional stress response observed in parental animals and could potentially prime offspring to respond to the same stress (*Burton et al., 2020*). Here, we exposed *C. elegans*, *C. briggsae*, *C. kamaaina*, and *C. tropicalis* to either osmotic stress or *P. vranovensis* infection and subsequently performed RNA-seq on offspring to test: (1) if the specific heritable changes in gene expression in response to each of these stresses are conserved across species and (2) if any changes in gene expression correlate with the phenotypic differences in intergenerational responses to stress we observed in the different species. This analysis allowed us to compare the effects of parental stress on offspring gene expression of 7587 single-copy orthologs that are conserved across all four species (*Supplementary file 1*).

Consistent with previous observations in *C. elegans*, we found that parental exposure to *P. vranovensis* resulted in substantial changes in offspring gene expression in all four species we investigated (*Figure 2* and *Supplementary file 2*). Of the 7587 single-copy orthologs shared between the four species, we identified 367 genes that exhibited a greater than twofold change in expression in the offspring of infected animals in *C. elegans* (*padj* <0.01) and at least one other species (*Figure 2D* and *Supplementary file 2*). Furthermore, we found that 35 genes exhibited a greater than twofold change in expression (*padj* <0.01) in the offspring of parents exposed to *P. vranovensis* in all four species (*Figure 2D* and *Table 1*). These data indicate that parental exposure to the bacterial pathogen *P. vranovensis* leads to changes in offspring gene expression at a common set of stress–response genes in diverse species of *Caenorhabditis*.

We performed the same analysis on the offspring of all four species from parents exposed to osmotic stress. From this analysis, we observed that parental exposure to osmotic stress resulted in 235 genes exhibiting differential expression in both *C. elegans* and *C. briggsae* offspring (*Figure 2F–J* and *Supplementary file 3*). In addition, we found that these changes in gene expression were largely distinct from the gene expression changes observed in the offspring of parents exposed to *P. vranovensis* (*Figure 2K* and *Supplementary files 2 and 3*), indicating that different parental stresses have distinct effects on offspring gene expression. However, parental exposure to *C. kamaaina* and *C. tropicalis* to osmotic stress resulted in approximately fivefold fewer changes in offspring gene expression (*Figure 2G–H* and *Supplementary file 3*). In total five genes (*C30B5.6*, *hphd-1*, *C42D4.3*, *ttr-15*, and *F08F3.4*) exhibited differential expression in the offspring of parents exposed to osmotic stress in all four species (*Figure 2I* and *Table 1*) and three of these five (*C30B5.6*, *hphd-1*, and *C42D4.3*) were also observed to change in the offspring of parents exposed to *P. vranovensis* (*Table 1*).

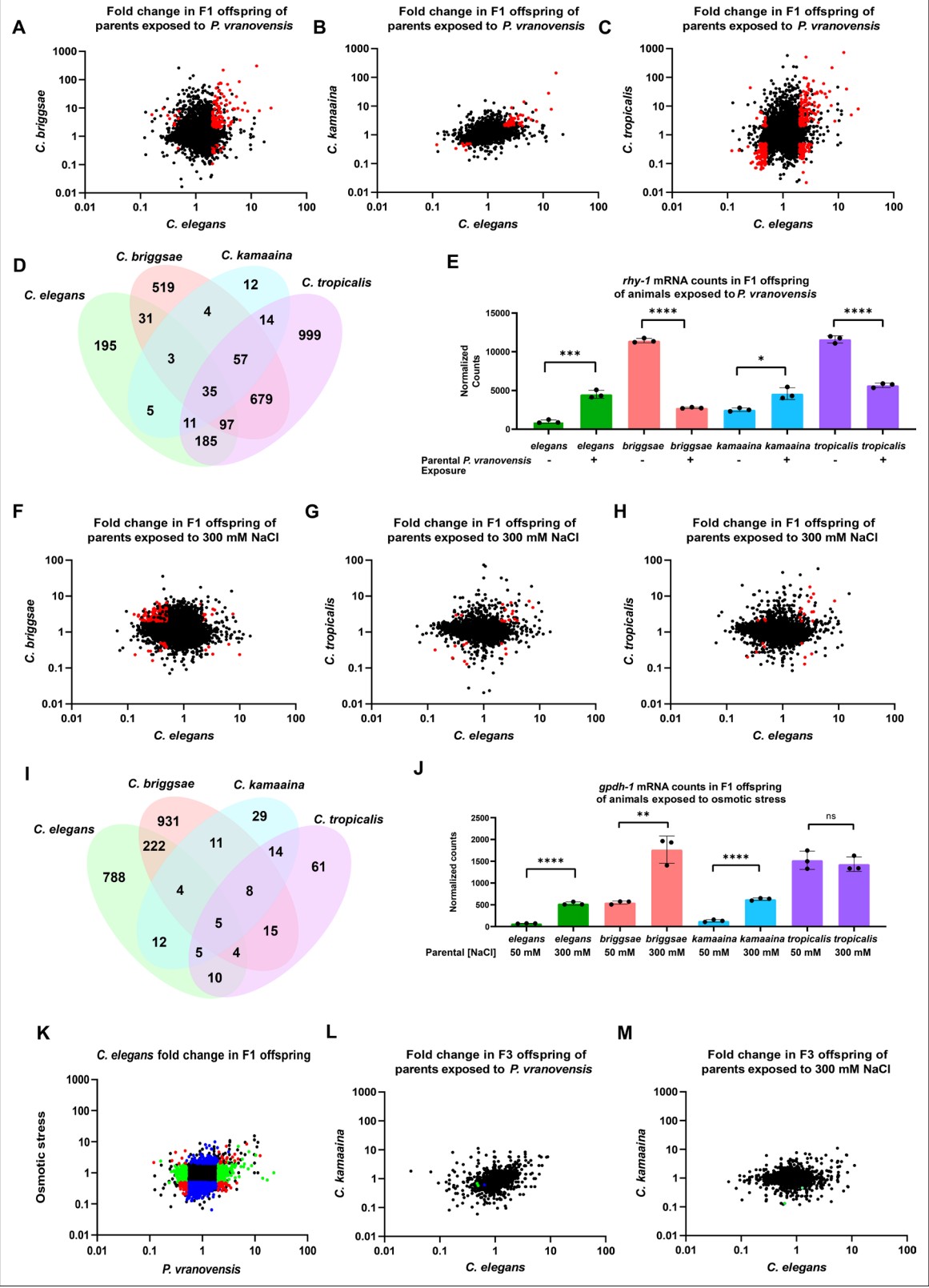

**Figure 2.** Parental exposure to *P. vranovensis* and osmotic stress have overlapping effects on offspring gene expression across multiple species. (**A**) Average fold change of 7587 single-copy ortholog genes in F1 progeny of *C. elegans* and *C. briggsae* parents fed *P. vranovensis* BIGb0446 when compared to parents fed *E. coli* HB101. Average fold change from three replicates. Red dots represent genes that exhibit > twofold (*padj* <0.01) changes in expression in both species. (**B**) Average fold change of 7587 single-copy ortholog genes in F1 progeny of *C. elegans* and *C. kamaaina*

*Figure 2 continued on next page*

Figure 2 continued

parents fed *P. vranovensis* BIGb0446 when compared to parents fed *E. coli* HB101. Average fold change from three replicates. Red dots represent genes that exhibit > twofold (*padj* <0.01) changes in expression in both species. (**C**) Average fold change of 7587 single-copy ortholog genes in F1 progeny of *C. elegans* and *C. tropicalis* parents fed *P. vranovensis* BIGb0446 when compared to parents fed *E. coli* HB101. Average fold change from three replicates. Red dots represent genes that exhibit > twofold (*padj* <0.01) changes in expression in both species (**D**) Venn diagram of the number of genes that exhibit overlapping >2 fold (*padj* <0.01) changes in expression in F1 progeny of animals exposed to *P. vranovensis* BIGb0446 in each species. (**E**) Normalized counts of reads matching orthologs of *rhy-1* in the F1 offspring of parents fed either *E. coli* HB101 or *P. vranovensis* BIGb0446. Data from **Supplementary file 2**. *n* = 3 replicates. (**F**) Average fold change of 7587 single-copy ortholog genes in F1 progeny of *C. elegans* and *C. briggsae* parents grown at 300 mM NaCl when compared to parents grown at 50 mM NaCl. Average fold change from three replicates. Red dots represent genes that exhibit > twofold (*padj* <0.01) changes in expression in both species. (**G**) Average fold change of 7587 single-copy ortholog genes in F1 progeny of *C. elegans* and *C. kamaaina* parents grown at 300 mM NaCl when compared to parents grown at 50 mM NaCl. Average fold change from three replicates. Red dots represent genes that exhibit > twofold (*padj* <0.01) changes in expression in both species in both species. (**H**) Average fold change of 7587 single-copy ortholog genes in F1 progeny of *C. elegans* and *C. tropicalis* parents grown at 300 mM NaCl when compared to parents grown at 50 mM NaCl. Average fold change from three replicates. Red dots represent genes exhibit > twofold (*padj* <0.01) changes in expression in both species. (**I**) Venn diagram of the number of genes that exhibit overlapping > twofold (*padj* <0.01) changes in expression in F1 progeny of animals grown at 300 mM NaCl in each species. (**J**) Normalized counts of reads matching orthologs of *gpdh-1* in the F1 progeny of parents grown at either 300 mM NaCl or 50 mM NaCl. Data from **Supplementary file 3**. *n* = 3 replicates. (**K**) Average fold change for 7587 ortholog genes in F1 progeny of *C. elegans* parents fed *P. vranovensis* or exposed to 300 mM NaCl when compared to naive parents. Average fold change from three replicates. Red dots – genes that change in expression in response to both stresses. Blue dots – genes that change in expression in response to only osmotic stress. Green dots – genes that change in expression in response to only *P. vranovensis*. (**L**) Average fold change of 7512 single-copy ortholog genes in F3 progeny of *C. elegans* and *C. kamaaina* fed *P. vranovensis* BIGb0446 when compared to those fed *E. coli* HB101. Average fold change from three replicates. Blue dots represent genes that exhibited > twofold (*padj* <0.01) changes in expression in *C. elegans*. Green dots represent genes that exhibited > twofold (*padj* <0.01) changes in expression in *C. kamaaina*. (**M**) Average fold change of 7512 single-copy ortholog genes in F1 progeny of *C. elegans* and *C. kamaaina* parents grown at 300 mM NaCl when compared to parents grown at 50 mM NaCl. Average fold change from three replicates. Green dots represent genes that exhibited > twofold (*padj* <0.01) changes in expression in *C. kamaaina*. *p < 0.05, **p < 0.01, ***p < 0.0001, ****p < 0.0001.

The online version of this article includes the following figure supplement(s) for figure 2:

**Figure supplement 1.** Differences in developmental timing are insufficient to explain a majority of the observed differences in gene expression in the offspring of stressed parents.

## Pairing gene expression and phenotypic data across species significantly enriches for genes required for intergenerational adaptations

To further probe how parental exposure to environmental stresses affects offspring gene expression, we first analyzed the gene ontology of the 37 genes that exhibit changes in expression in the offspring of stressed parents in all four species using g:Profiler (*Raudvere et al., 2019*). We found that these 37 genes were significantly enriched for extracellular proteins ($p < 2.278 \times 10^{-3}$). However, no additional commonalities were identified and none of these 37 genes have previously been linked to adaptations to *P. vranovensis* infection or osmotic stress.

We found that different species exhibit different intergenerational responses to both *P. vranovensis* infection and osmotic stress (*Figure 1*). We hypothesized that the effects of parental exposure to environmental stresses on offspring gene expression might correlate with how offspring phenotypically respond to stress. Parental exposure of *C. elegans* and *C. kamaaina* to *P. vranovensis* led to increased progeny resistance to future *P. vranovensis* exposure (*Figure 1B*). By contrast, parental exposure of *C. briggsae* to *P. vranovensis* led to increased offspring susceptibility to *P. vranovensis* (*Figure 1B*). We hypothesized that differences in the expression of genes previously reported to be required for adaptation to *P. vranovensis*, such as the acyltransferase *rhy-1*, might underlie these differences between species. We therefore investigated whether any genes exhibited specific changes in expression in *C. elegans* and *C. kamaaina* that were either absent or inverted in *C. briggsae*. We found that of the 562 genes that exhibited a greater than twofold change in expression in the offspring of parents exposed to *P. vranovensis* in *C. elegans*, only 54 also exhibited a greater than twofold intergenerational change in expression in *C. kamaaina* (*Supplementary file 2*). From this refined list of 54 genes, 17 genes either did not exhibit a change in *C. briggsae* or changed in the opposite direction (*Table 2*).

Consistent with our hypothesis that intergenerational gene expression changes across species might correlate with their phenotypic responses, we found that all three genes previously reported to be required for the intergenerational adaptation to *P. vranovensis* (*rhy-1*, *cysl-1*, and *cysl-2* – *Burton et al., 2020*) were among the 17 genes that exhibited differential expression in *C. elegans* and *C.*

**Table 1.** Complete list of genes that exhibited a greater than twofold change in expression in the F1 progeny of parents exposed to *P. vranovensis* or osmotic stress in all four species tested.

| Genes that change in F1 progeny of all species exposed to *P. vranovensis* | Predicted function |
|---|---|
| C18A11.1 | Unknown |
| R13A1.5 | Unknown |
| D1053.3 | Unknown |
| pmp-5 | ATP-binding activity and ATPase-coupled transmembrane transporter activity, ortholog of human ABCD4 |
| C39E9.8 | Unknown |
| nit-1 | Nitrilase ortholog – predicted to enable hydrolase activity |
| lips-10 | Lipase related |
| srr-6 | Serpentine receptor, class R |
| Y51B9A.6 | Predicted to enable transmembrane transporter activity |
| gst-33 | Glutathione *S*-transferase |
| ptr-8 | Patched domain containing, ortholog of human PTCHD1, PTCHD3, and PTCHD4 |
| ZC443.1 | Predicted to enable D-threo-aldose 1-dehydrogenase activity |
| cri-2 | Conserved regulator of innate immunity, ortholog of human TIMP2 |
| Y42G9A.3 | Unknown |
| ttr-21 | Transthyretin-related, involved in response to Gram-negative bacteria |
| F45E4.5 | Involved in defense against Gram-negative bacteria |
| C42D4.1 | Domain of unknown function DUF148 |
| asp-14 | Predicted to enable aspartic-type endopeptidase activity. Involved in innate immune response |
| cyp-32B1 | Cytochrome P450 family. Ortholog of human CYP4V2 |
| nas-10 | Predicted to enable metalloendopeptidase activity and zinc ion-binding activity |
| W01F3.2 | Unknown |
| nhr-11 | Nuclear hormone receptor |
| F26G1.2 | Unknown |
| F48E3.2 | Predicted to enable transmembrane transporter activity |
| hpo-26 | Unknown, hypersensitive to pore forming toxin |
| R05H10.1 | Unknown |
| C08E8.4 | Involved in innate immune response |
| C11G10.1 | Unknown |
| Y73F4A.2 | Unknown, DOMON domain containing |
| bigr-1 | Predicted to enable hydrolase activity |
| nlp-33 | Neuropeptide like, involved in innate immune response |
| far-3 | Predicted to enable lipid-binding activity |
| **Genes that change in F1 progeny of all species exposed to both osmotic stress and *P. vranovensis*** | |
| C30B5.6 | Unknown |
| hphd-1 | Predicted to enable hydroxyacid–oxoacid transhydrogenase activity. Ortholog of human ADHFE1 |
| C42D4.3 | Unknown, DB module and domain of unknown function DB |
| **Genes that change in F1 progeny of all species exposed to osmotic stress** | |
| ttr-15 | Transthyretin-like family |

*Table 1 continued on next page*

*Table 1 continued*

| Genes that change in F1 progeny of all species exposed to *P. vranovensis* | Predicted function |
|---|---|
| *F08F3.4* | Predicted to enable catalytic activity. Involved in innate immune response. Ortholog of human TDH |

**Table 2.** Complete list of genes that exhibited a consistent and greater than twofold change in expression in the F1 progeny of parents exposed to *P. vranovensis* or osmotic stress in only species that intergenerationally adapted to stress.

Genes listed for *P. vranovensis* were identified by comparing genes that change consistently in *C. elegans* and *C. kamaaina*, but not *C. briggsae*. Genes listed for osmotic stress were identified by comparing genes that change consistently in *C. elegans*, *C. briggsae*, and *C. kamaaina*, but not in *C. tropicalis*. Bold font indicates genes that have already been demonstrated to be involved in *C. elegans* adaption to these stresses.

| Genes that change consistently in F1 progeny of only species that adapt to *P. vranovensis* | Predicted function |
|---|---|
| *daf-18* | Lipid phosphatase, homologous to human PTEN tumor suppressor |
| *gst-38* | Glutathione *S*-transferase |
| *H04M03.3* | Predicted to enable oxidoreductase activity. |
| *oops-1* | Oocyte partner of SPE-11 |
| *F09G8.10* | Unknown |
| *glb-1* | Globin -related |
| *F57H12.6* | Unknown |
| *elo-6* | Predicted to enable transferase activity, transferring acyl groups, ortholog of human ELOVL3 and ELOVL6 |
| *cpr-5* | Predicted to enable cysteine-type peptidase activity |
| *xpo-2* | Exportin involved in nuclear export, ortholog of human CSE1L |
| **cysl-1** | Cysteine synthase known to be involved in adaptation to *P. vranovensis* |
| **rhy-1** | Regulator of HIF-1 known to be involved in adaptation to *P. vranovensis* |
| *cdc-25.1* | Homolog of human CDC25 phosphatase |
| *imb-1* | Importin beta family, ortholog of human KPNB1 |
| *VZK882L.2* | Unknown |
| **cysl-2** | Cysteine synthase known to be involved in adaptation to *P. vranovensis* |
| *cyk-7* | Involved in intercellular bridge organization |
| **Genes that change consistently in F1 progeny of only species that adapt to osmotic stress** | |
| *T05F1.9* | Unknown |
| *grl-21* | Unknown, Ground-like domain containing |
| *gpdh-1* | Glycerol-3-phosphate dehydrogenase known to be involved in osmotic stress resposne |
| *T22B7.3* | Amidinotransferase, ortholog of human DDAH1 and DDAH2 |

*kamaaina* but not in *C. briggsae*. This overlap is significantly above what is expected by chance (*P* < 1.337e−−08 – hypergeometric probability). We conclude that the effects of parental exposure to *P. vranovensis* on offspring gene expression correlate with their phenotypic response. Furthermore, we propose that this new list of 17 genes (*Table 2*) is likely to be enriched in additional conserved genes required for this intergenerational response to pathogen infection. This list includes multiple highly conserved genes including multiple factors involved in nuclear transport (*imb-1* and *xpo-2*), the CDC25 phosphatase ortholog *cdc-25.1*, and the PTEN tumor suppressor ortholog *daf-18*.

Notably, of the revised list of 17 genes, we identified a single gene that exhibited a greater than twofold increase in expression in *C. elegans* and *C. kamaaina* F1 progeny but had an inverted greater than twofold decrease in expression in *C. briggsae* F1 progeny. That gene is *rhy-1* (*Figure 2E*), one of the three genes known to be required for animals to intergenerationally adapt to *P. vranovensis* infection (*Burton et al., 2020*). This directional change of *rhy-1* expression in progeny of animals exposed to *P. vranovensis* correlates with the observation that parental exposure to *P. vranovensis* results in enhanced pathogen resistance in offspring in *C. elegans* and *C. kamaaina* but has a strong deleterious effect on pathogen resistance in *C. briggsae* (*Figure 1B*). Collectively, these findings suggest that molecular mechanisms underlying adaptive and deleterious effects in different species might be related and dependent on the direction of changes in gene expression of specific stress–response genes.

We performed the same analysis pairing our transcriptional data with our phenotypic data for the intergenerational response to osmotic stress. We found that *C. elegans, C. briggsae,* and *C. kamaaina* intergenerationally adapted to osmotic stress, but *C. tropicalis* did not (*Figure 1D*). We therefore identified genes that were differentially expressed in the F1 offspring of *C. elegans, C. briggsae,* and *C. kamaaina* exposed to osmotic stress, but not in *C. tropicalis.* From this analysis, we identified four genes (*T05F1.9, grl-21, gpdh-1,* and *T22B7.3*) that are specifically differentially expressed in the three species that adapt to osmotic stress but not in *C. tropicalis* (*Table 2*); this list of genes includes the glycerol-3-phosphate dehydrogenase *gpdh-1* which is one of the most upregulated genes in response to osmotic stress and is known to be required for animals to properly respond to osmotic stress (*Lamitina et al., 2006*). These results suggest that, similar to our observations for *P. vranovensis* infection, different patterns in the expression of known osmotic stress response genes correlate with different intergenerational phenotypic responses to osmotic stress.

Differences in gene expression in the offspring of stressed parents could be due to programmed changes in expression in response to stress or due to indirect effects caused by changes in developmental timing. To confirm that the embryos from all conditions were collected at the same developmental stage we compared our RNA-seq findings to a time-resolved transcriptome of *C. elegans* development (*Boeck et al., 2016*). Consistent with our visual observations that a vast majority of offspring collected were in the comma stage of embryo development, we found that the gene expression profiles of all offspring from both naive and stressed parents overlapped strongly with the 330- to 450- min time points of development (*Figure 2—figure supplement 1*). In addition, we found that approximately 50 % of all genes that were differentially expressed in the offspring of stressed parents when compared to naive parents exhibited a change in gene expression that was more than one standard deviation outside their average expression across all time points of embryo development (*Figure 2—figure supplement 1B-C*). We similarly found that many of the genes known to be required for intergenerational responses to stress exhibit expression that is outside the range of expression observed at any time point of early development (*Figure 2—figure supplement 1D-E*). These results suggest that a majority of the expression differences we observed in the offspring of stressed parents were not due to differences in developmental timing.

## The effects of parental bacterial infection and osmotic stress on offspring gene expression are not maintained transgenerationally

Determining whether the effects of parental exposure to stress on offspring gene expression are reversible after one generation or if any changes in gene expression persist transgenerationally is a critical and largely unanswered question in the field of multigenerational effects. To test if any of the intergenerational changes in gene expression that we observed persist transgenerationally, we performed RNA-seq of F3 progeny of *C. elegans* exposed to both *P. vranovensis* and osmotic stress. We found that none of the 1515 genes that exhibited differential expression in F1 progeny

for either *P. vranovensis* infection or osmotic stress were also differentially expressed in *C. elegans* F3 progeny (*Figure 2L and M* and *Supplementary file 4*). We conclude that, at minimum, the vast majority of intergenerational effects of these stresses on gene expression in *C. elegans* do not persist transgenerationally.

We hypothesized that transgenerational effects on gene expression could potentially be more robust in other species. We therefore performed the same analysis on F3 gene expression in response to both *P. vranovensis* infection and osmotic stress in a second species that intergenerationally adapts to both stresses, *C. kamaaina*. We again found that none of the genes that exhibited differential expression in F1 progeny of parents exposed to *P. vranovensis* were also differentially expressed in F3 progeny (*Figure 2L* and *Supplementary file 4*). We did, however, identify two genes, the *C. kamaaina* orthologs of *C. elegans hphd-1* and *C09B8.4*, that exhibited differential expression in both the F1 and F3 progeny of parents exposed to osmotic stress (*Figure 2M* and *Supplementary file 4*). It is possible that these two genes represent true transgenerational effects on gene expression, but given that these effects were not also observed in *C. elegans* and that only two genes were identified out of thousands of possible gene comparisons using a false discovery cutoff of 1 %, we cannot rule out that these two genes are false positives. Collectively, our results suggest that neither of these biotic or abiotic stresses that elicit robust intergenerational changes in gene expression cause similar transgenerational changes in gene expression under the same conditions in multiple different species. We note, however, that it remains possible that transgenerational effects of these stresses could persist through other mechanisms, could affect the expression of genes that are not clearly conserved between species, or could exert weaker effects on broad classes of genes that would not be detectable at any specific individual loci as was reported for the transgenerational effects of starvation and loss of COMPASS complex function on gene expression in *C. elegans* (*Greer et al., 2011*; *Webster et al., 2018*). Furthermore, it is possible that transgenerational effects on gene expression in *C. elegans* are restricted to germ cells (*Buckley et al., 2012*; *Houri-Zeevi et al., 2020*; *Posner et al., 2019*) or to a small number of cells and are not detectable when profiling gene expression in somatic tissue from whole animals.

## Intergenerational responses to stress can have deleterious tradeoffs

Intergenerational changes in animal physiology that protect offspring from future exposure to stress could be stress-specific or could converge on a broadly stress-resistant state. If intergenerational adaptive effects are stress-specific, then it is expected that parental exposure to a given stress will protect offspring from that same stress but potentially come at the expense of fitness in mismatched environments. If intergenerational adaptations to stress converge on a generally more stress-resistant state, then parental exposure to one stress might protect offspring against many different types of stress. To determine if the intergenerational effects we investigated here represent specific or general responses, we assayed how parental *C. elegans* exposure to osmotic stress, *P. vranovensis* infection, and *N. parisii* infection, either alone or in combination, affected offspring responses to mismatched stresses. We found that parental exposure to *P. vranovensis* did not affect the ability of animals to intergenerationally adapt to osmotic stress (*Figure 3A*). By contrast, parental exposure to osmotic stress completely eliminated the ability of animals to intergenerationally adapt to *P. vranovensis* (*Figure 3B*). This effect is unlikely to be due to the effects of osmotic stress on *P. vranovensis* itself, as mutant animals that constitutively activate the osmotic stress response (*osm-8*) were also completely unable to adapt to *P. vranovensis* infection (*Figure 3C*; *Rohlfing et al., 2011*). We conclude that animals' intergenerational responses to *P. vranovensis* and osmotic stress are stress-specific, consistent with our observation that parental exposure to these two stresses resulted in distinct changes in offspring gene expression (*Figure 2K*).

We performed a similar analysis comparing animals' intergenerational response to osmotic stress and the eukaryotic pathogen *N. parisii*. We previously reported that L1 parental infection with *N. parisii* results in progeny that is more sensitive to osmotic stress (*Willis et al., 2021*). Here, we found that L4 parental exposure of *C. elegans* to *N. parisii* had a small, but not significant effect on offspring response to osmotic stress (*Figure 3D*). However, similar to our observations for osmotic stress and bacterial infection, we found that parental exposure to both osmotic stress and *N. parisii* infection simultaneously resulted in offspring that were less protected against future *N. parisii* infection than when parents are exposed to *N. parisii* alone (*Figure 3E*). Collectively, these data further support the

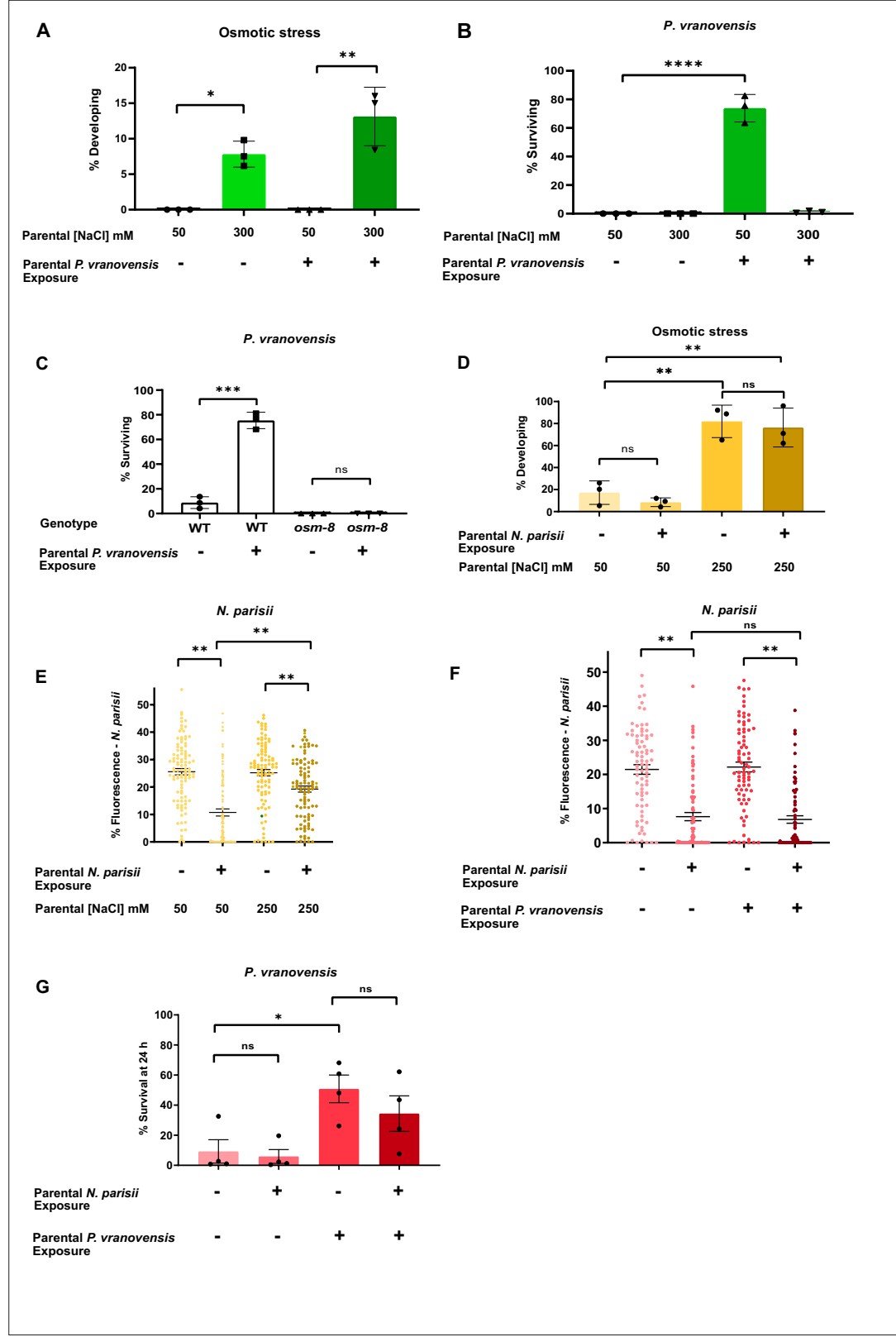

**Figure 3.** Intergenerational adaptations to stress are stress-specific and have deleterious tradeoffs. (**A**) Percent of wild-type *C. elegans* mobile and developing at 500 mM NaCl after 24 hr. Data presented as mean values ± s.d. *n* = 3 experiments of >100 animals. (**B**) Percent of wild-type *C. elegans* surviving after 24 hr of exposure to *P. vranovensis* BIGb0446. Data presented as mean values ± s.d. *n* = 3 experiments of >100 animals. (**C**) Percent

*Figure 3 continued on next page*

*Figure 3 continued*

of wild-type and *osm-8(n1518) C. elegans* surviving after 24 hr of exposure to *P. vranovensis* BIGb0446. Data presented as mean values ± s.d. *n* = 3 experiments of >100 animals. (**D**) Percent of wild-type *C. elegans* mobile and developing at 420 mM NaCl after 48 hr. Data presented as mean values ± s.d. *n* = 3 experiments of >100 animals. (**E**) *N. parisii* parasite burden of individual *C. elegans* after 72 hr (as determined by percentage fluorescence from DY96-stained spores after 72 hr). Data presented as mean values ± s.e.m. *n* = 4 experiments of 25 animals (**F**) *N. parisii* parasite burden of individual *C. elegans* after 72 hr (as determined by percentage fluorescence from DY96-stained spores after 72 hr). Data presented as mean values ± s.e.m. *n* = 3 experiments of 25 animals. (**G**) Percent of wild-type *C. elegans* surviving after 24 hr of exposure to *P. vranovensis* BIGb0446. Data presented as mean values ± s.e.m. *n* = 3 experiments of >100 animals. *p < 0.05, **p < 0.01, ***p < 0.0001, ****p < 0.0001.

The online version of this article includes the following figure supplement(s) for figure 3:

**Source data 1.** Statistics source data for *Figure 3*.

conclusion that intergenerational responses to infection and osmotic stress are stress-specific and suggest that intergenerational adaptations to osmotic stress might come at the expense of animals' ability to properly respond to bacterial or eukaryotic infections when either is paired with osmotic stress.

To compare animals' intergenerational responses to bacterial infection and eukaryotic infection, we performed a similar comparative analysis. We found that parental exposure to *P. vranovnesis* had no observable effect on offspring response to *N. parisii* either alone or when both pathogens were present simultaneously (*Figure 1F*). Similarly, we found that parental exposure to *N. parisii* had no observable effect on offspring response to *P. vranovensis* either alone or when both pathogens were present at the same time (*Figure 1G*). We conclude that intergenerational adaptations to osmotic stress, *P. vranovensis* infection and *N. parisii* infection are largely stress-specific.

## Intergenerational responses to *Pseudomonas* pathogens are distinct from other bacterial pathogens

To further probe the specificity of intergenerational responses to stress, we also sought to determine if the substantial changes in pathogen resistance and gene expression observed in *C. elegans* offspring from parents exposed to the bacterial pathogen *P. vranovensis* were specific to this pathogen or part of a general response to bacterial pathogens. We previously found that the transcriptional response to *P. vranovensis* in F1 progeny is distinct from the response to *P. aeruginosa* (*Burton et al., 2020*). To further probe the specificity of this intergenerational response, we first screened wild bacterial isolates from France (*Samuel et al., 2016*) and the United Kingdom (*Supplementary file 5*) for those that are potential natural pathogens of *C. elegans* and that also intergenerationally affect *C. elegans* survival or growth rate. From this analysis, we identified a new *Pseudomonas* isolate, *Pseudomonas* sp. 15C5, where parental exposure to *Pseudomonas* sp. 15C5 enhanced offspring growth rate in response to future exposure to *Pseudomonas* sp. 15C5 (*Figure 4A*). This intergenerational effect resembled *C. elegans* intergenerational adaptation to *P. vranovensis* and we found that parental exposure to either isolate of *Pseudomonas* protected offspring from future exposure to the other *Pseudomonas* isolate (*Figure 4A–B*). To test if *Pseudomonas* sp. *15* C5 was a new isolate of *P. vranovensis* or a distinct species of *Pseudomonas,* we performed both 16 S rRNA sequencing and sequenced the gene *rpoD* of *Pseudomonas* sp. 15C5. From this analysis, we found that *Pseudomonas* sp. 15C5 is not an isolate of *P. vranovensis* and is most similar to *Pseudomonas putida* – 99.49 % identical 16 S rRNA and 98.86 % identical *rpoD* by BLAST (*Supplementary file 6*). These results indicate that parental exposure to multiple different *Pseudomonas* species can protect offspring from future pathogen exposure. We note, however, that other pathogenic species of *Pseudomonas*, such as *P. aeruginosa*, did not cross protect against *P. vranovensis* (*Burton et al., 2020*), indicating that not all pathogenic species of *Pseudomonas* result in the same intergenerational in offspring pathogen resistance.

In addition to these intergenerational adaptive effects, we also identified two bacterial isolates that activate pathogen–response pathways, *Serretia plymutica* BUR1537 and *Aeromonas* sp. BIGb0469 (*Samuel et al., 2016*; *Hellberg et al., 2015*), that resulted in intergenerational deleterious effects (*Figure 4C–D*). Parental exposure of animals to these potential bacterial pathogens did not intergenerationally protect animals against *P. vranovensis* (*Figure 4—figure supplement 1*). We conclude that parental exposure to some species of *Pseudomonas* can protect offspring from other species of

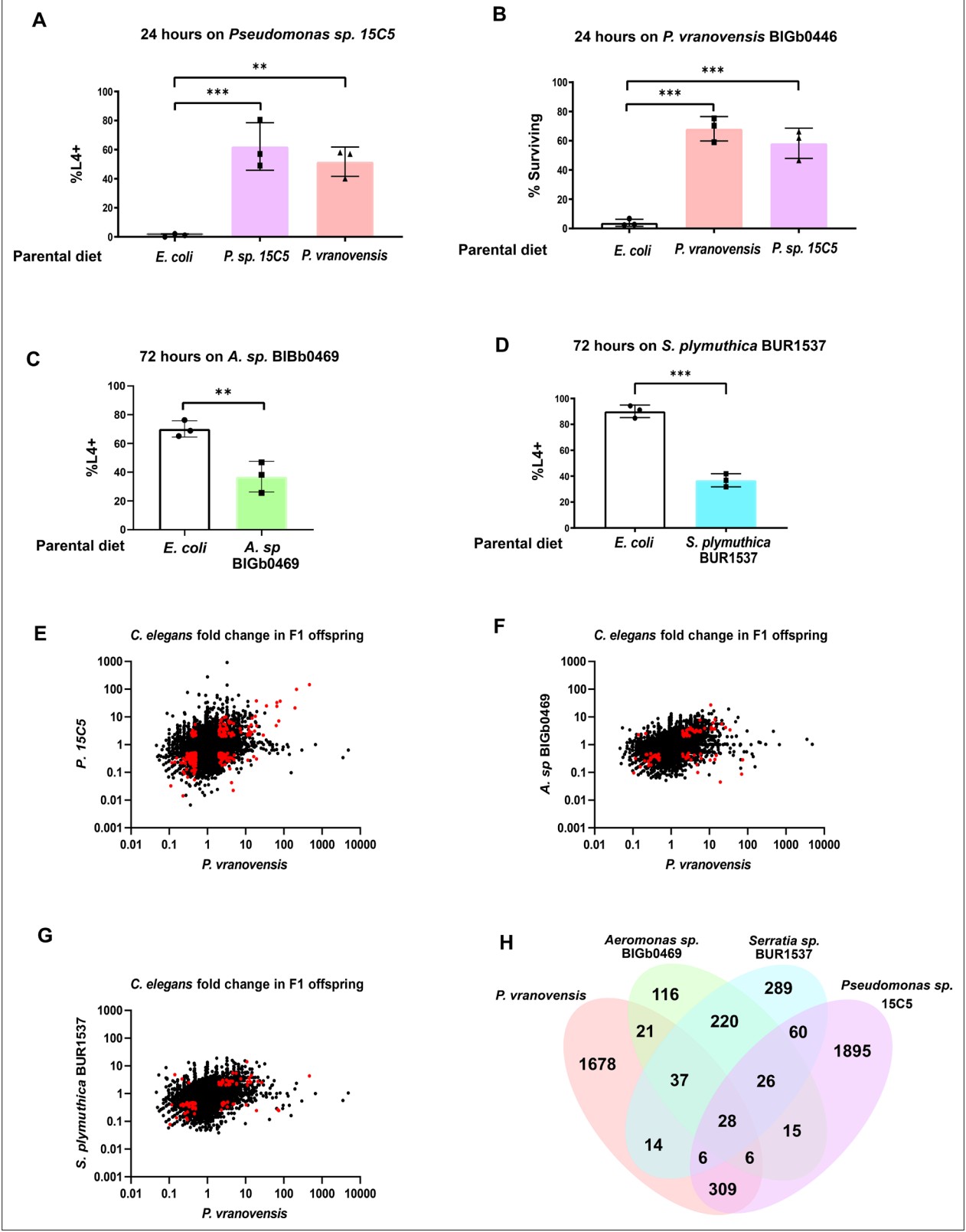

**Figure 4.** Many of the intergenerational effects of parental exposure to bacterial pathogens on offspring gene expression are pathogen specific. (**A**) Percent of wild-type *C. elegans* that developed to the L4 larval stage after 48 hr of feeding on *Pseudomonas* sp. 15C5. Data presented as mean values ± s.d. *n* = 3 experiments of >100 animals. (**B**) Percent of wild-type *C. elegans* surviving after 24 hr of exposure to *P. vranovensis* BIGb0446. Data presented as mean values ± s.d. *n* = 3 experiments of >100 animals. (**C**) Percent of wild-type *C. elegans* that developed to the L4 larval stage after 48 hr

*Figure 4 continued*

of feeding on *Aeromonas* sp. BIGb0469. Data presented as mean values ± s.d. *n* = 3 experiments of >100 animals. (**D**) Percent of wild-type *C. elegans* that developed to the L4 larval stage after 48 hr of feeding on *Serratia plymuthica* BUR1537. Data presented as mean values ± s.d. *n* = 3 experiments of >100 animals. (**E**) Average fold change of genes in F1 progeny of *C. elegans* fed either *Pseudomonas* sp. 15C5 or *P. vranovensis* BIGb0446 when compared to parents fed *E. coli* HB101. Average fold change from three replicates. Red dots represent genes that exhibit statistically significant (*padj* <0.01) changes in the F1 offspring of parents fed both *Pseudomonas* sp. 15C5 and *P. vranovensis* BIGb0446. (**F**) Average fold change of genes in F1 progeny of *C. elegans* fed either *Aeromonas* sp. BIGb0469 or *P. vranovensis* BIGb0446 when compared to parents fed *E. coli* HB101. Average fold change from three replicates. Red dots represent genes that exhibit statistically significant (*padj* <0.01) changes in the F1 offspring of parents fed both *Aeromonas* sp. BIGb0469 and *P. vranovensis* BIGb0446. (**G**) Average fold change of genes in F1 progeny of *C. elegans* fed either *S. plymuthica* BUR1537 or *P. vranovensis* BIGb0446 when compared to parents fed *E. coli* HB101. Average fold change from three replicates. Red dots represent genes that exhibit statistically significant (*padj* <0.01) changes in the F1 offspring of parents fed both *S. plymuthica* BUR1537 and *P. vranovensis* BIGb0446. (**H**) Venn diagram of the number of genes that exhibit overlapping statistically significant (*padj* <0.01) changes in expression in F1 progeny of *C. elegans* parents fed each different bacterial species. **p < 0.01, ***p < 0.0001.

The online version of this article includes the following figure supplement(s) for figure 4:

**Source data 1.** Statistics source data for *Figure 4*.

**Figure supplement 1.** Parental exposure to *Aeromonas* sp. BIGb0469 and *S. plymuthica* BUR1537 does not protect offspring from *P. vranovensis.*

*Pseudomonas*, but that these effects are likely specific to a subset of *Pseudomonas* species and not part of a broad response to Gram-negative bacterial pathogens.

To determine how different parental bacterial infections affect offspring gene expression patterns, we profiled gene expression in the offspring of *C. elegans* parents exposed to each of *P. vranovensis* BIGb0427, *Pseudomonas* sp. 15C5, *Serretia plymuthica* BUR1537, and *Aeromonas* sp. BIGb0469. We found that only 28 genes exhibit differential expression in the offspring of parents exposed to all four potential pathogens (*Figure 4E–H*). However, we identified 309 genes that are specifically differentially expressed in the offspring of parents exposed to *P. vranovensis* and *Pseudomonas* sp. 15C5 but not in the offspring of parents exposed to *S. plymuthica* BUR1537 or *Aeromonas* sp. BIGb0469 (*Figure 4H* and *Supplementary file 7*). We conclude that parental exposure to bacterial pathogens that elicit enhanced offspring resistance to *P. vranovensis* resulted in distinct changes in offspring gene expression that are not observed when parents are exposed to other Gram-negative bacterial pathogens. Collectively, our results suggest that a majority of the intergenerational effects of a parent's environment on offspring gene expression are both stress and pathogen-specific.

## Discussion

Overall, our findings support the conclusion that some types of intergenerational effects are conserved while others diverge, not unlike other important aspects of biology that vary between the species investigated here such as mode of reproduction – *C. elegans*, *C. briggsae*, and *C. tropicalis* are hermaphroditic while *C. kamaaina* is an exclusively male–female species. Specifically, our findings provide some of the first evidence that the mechanisms underlying intergenerational effects of a parent's environment on offspring are evolutionarily conserved among different species. While these findings are restricted to the genus *Caenorhabditis*, to our knowledge they represent the first observation that intergenerational responses to stress are conserved across any evolutionary distance, and they provide a base from which we can compare the numerous different reported observations of multigenerational effects in *C. elegans* to similar intergenerational responses to stress in more distantly related species. For example, we found that only a subset of the many transcriptional changes that are detectable in the offspring of stressed *C. elegans* parents are conserved in any other species investigated. In addition, we used our analysis to identify 37 genes that exhibited intergenerational regulation of expression in response to the specific stresses of *P. vranovensis* infection or osmotic stress in all species studied (*Figure 3*). We propose that these genes might be particularly tuned for intergenerational regulation and might similarly be involved in intergenerational responses to stress in more distantly related species, including species outside the *Caenorhabditis* genus.

Notably, we found that the expression of these 37 genes in the offspring of parents exposed to either *P. vranovensis* infection or osmotic stress were still differentially expressed in *C. tropicalis* even though parental exposure to these stresses did not appear to affect offspring stress resistance in either assay (*Figures 1 and 2*). We hypothesize that the molecular consequences of parental stress

on offspring, such as changes in the expression of stress–response genes, might be more easily identifiable than the specific physiological consequences of parental stress on offspring. In this case, we might not have detected the unique phenotypic effects of parental exposure to stress on offspring in *C. tropicalis* using our assay conditions, but such effects might still exist in this species and be related to those observed in other species. Future studies of the phenotypic effects of parental stress on offspring across species will likely shed significant light on how similar molecular mechanisms can mediate different intergenerational responses to stress across evolution.

Consistent with the hypothesis that parental exposure to the same stress might elicit distinct phenotypic effects on offspring in different species via evolutionarily related mechanisms, we found that parental exposure of *C. briggsae* to *P. vranovensis* had a strong deleterious effect on offspring pathogen resistance even though parental exposure of *C. elegans* and *C. kamaaina* to *P. vranovensis* resulted in increased offspring resistance to *P. vranovensis* (*Figure 1B*). This inversion of an intergenerational effect from a presumed adaptive effect to a presumed deleterious effect correlated with an inversion in the expression of specific pathogen–response genes that were previously reported to be required for animals to intergenerationally adapt to *P. vranovensis*, such as *rhy-1* which exhibits increased expression in *C. elegans* and *C. kamaaina* offspring from infected parents but decreased expression in *C. briggsae* offspring from infected parents (*Figure 2E*). To our knowledge, these findings are the first to suggest that the molecular mechanisms underlying presumed adaptive and deleterious intergenerational effects in different species are evolutionarily related at the gene expression level. These findings suggest that similar observations of presumed intergenerational deleterious effects in diverse species, such as fetal programming in humans, might also be molecularly related to intergenerational adaptive effects in other species. Alternatively, our findings suggest that presumed intergenerational deleterious effects might in fact represent deleterious tradeoffs that are adaptive in other contexts. We expect that a more complete consideration of the evolution of intergenerational effects and the potential relationship between adaptive and deleterious effects will play an important role in understanding how intergenerational effects contribute to organismal resilience in changing environments, what role such effects play in evolution, and how such effects contribute to multiple human pathologies associated with a parent's environment (*Langley-Evans, 2006*).

Lastly, the extent to which intergenerational and transgenerational responses to environmental stress represent related, independent, or even mutually exclusive phenomena represents a major outstanding question in the field of multigenerational effects. Evolutionary modeling of intergenerational and transgenerational effects has suggested that different ecological pressures favor the evolution of either intergenerational or transgenerational responses under different conditions. Specifically, it has been suggested that intergenerational effects are favored when offspring environmental conditions are predictable from the parental environment (*Dey et al., 2016*; *Lind et al., 2020*; *Proulx et al., 2019*; *Uller, 2008*). Furthermore, it has been speculated that intergenerational adaptations to stress will have costs (*Uller, 2008*). These costs, such as the costs we observed for animals intergenerational adaptation to osmotic stress (*Figure 3*), are likely to strongly favor the loss or active erasure of intergenerational effects if the parental environment improves to avoid potential deleterious effects when a stress is no longer present. By contrast, transgenerational effects were found to predominantly be favored when parental environmental cues are unreliable and the maintenance of information across many generations might be worth the potential costs (*Uller et al., 2015*).

Our findings in this study support either a model in which intergenerational and transgenerational effects represent potentially distinct phenomena or a model in which transgenerational effects only persist or occur under certain conditions with the vast majority of the effects of parental stress on offspring gene expression being lost or actively erased after one generation under other conditions. We strongly suspect that future studies into the mechanisms regulating these intergenerational effects will shed significant light on how intergenerational effects on gene expression are lost and/or erased. In addition, we expect that similar studies of transgenerational effects will potentially elucidate the circumstances under which animals decide if environmental information might be worth maintaining transgenerationally despite any potential tradeoffs and if the growing number of transgenerational effects observed in *C. elegans* are similarly evolutionarily conserved.

Lastly, future studies of intergenerational effects will be critical in determining the extent to which the mechanisms that mediate intergenerational effects are conserved outside of *Caenorhabditis* and if similar mechanisms to those uncovered in *C. elegans* mediate the numerous different adaptive and

deleterious intergenerational effects that have been reported in diverse taxa ranging from the intergenerational development of wings in aphids (*Vellichirammal et al., 2017*) to fetal programming and the role it plays in disease in humans (*Langley-Evans, 2006*).

# Materials and methods
## Strains
*C. elegans* strains were cultured and maintained at 20 °C unless noted otherwise. The Bristol strain N2 was the wild-type strain. Wild-isolate strains used in the main figures of this study: N2 (*C. elegans*), AF16 (*C. briggsae*), JU1373 (*C. tropicalis*), and QG122 (*C. kamaaina*). Wild-isolate strains used in figure supplements of this study: MY1 (*C. elegans*), PS2025 (*C. elegans*), CX11262 (*C. elegans*), JU440 (*C. elegans*), JU778 (*C. elegans*), JU1213 (*C. elegans*), LKC34 (*C. elegans*), JU1491 (*C. elegans*), EG4724 (*C. elegans*), KR314 (*C. elegans*), SX1125 (*C. briggsae*), and JU1348 (*C. briggsae*). Mutant alleles used in this study: *osm-8(n1518)* and *Cbr-gpdh-2(syb2973)*.

## *P. vranovensis* survival assays
*P. vranovensis* BIGb0446 or *Pseudomonas* sp. 15C5 was cultured in LB at 37 °C overnight. 1 ml of overnight culture was seeded onto 50 mm NGM agar plates and dried in a laminar flow hood (bacterial lawns completely covered the plate such that animals could not avoid the pathogen). All plates seeded with BIGb0446 or 15C5 were used the same day they were seeded. Young adult animals were placed onto 50 mm NGM agar plates seeded with 1 ml either *E. coli* HB101, *P. vranovensis* BIGb446, or *Pseudomonas* sp. 15C5 for 24 h at room temperature (22 °C). Embryos from these animals were collected by bleaching and placed onto fresh NGM agar plates seeded with BIGb0446. Percent surviving were counted after 24 hr at room temperature (22 °C) unless otherwise noted.

## Osmotic stress and *P. vranovensis* multiple stress adaptation assays
Young adult animals that were grown on NGM agar plates seeded with *E. coli* HB101 were collected and transferred to new 50 mM NaCl control plates seeded with *E. coli* HB101, 300 mM NaCl plates seeded with *E. coli* HB101, 50 mM NaCl control plates seeded with *P. vranovensis* BIGb0446, or 300 mM NaCl plates seeded with *P. vranovensis* BIGb0446. Animals were grown for 24 hr at room temperature (22 °C). Embryos from these animals were collected by bleaching and transferred to new 500 mM NaCl plates seeded with *E. coli* HB101 or 50 mM NaCl plates seeded with *P. vranovensis* BIGb0446. Percent of animals developing or surviving was scored after 24 hr at room temperature as previously described in *Burton et al., 2017* and *Burton et al., 2020*.

## Preparation of *N. parisii* spores
Spores were prepared as described previously (*Willis et al., 2021*). In brief, large populations of *C. elegans* N2 were infected with microsporidia spores. Infected worms were harvested and mechanically disrupted using 1 mm diameter Zirconia beads (BioSpec). Resulting lysate was filtered through 5 μm filters (Millipore Sigma) to remove nematode debris. Spore preparations were tested for contamination and those free of contaminating bacteria were stored at −80 °C.

## *N. parisii* infection assays and multiple stress adaptation assays
P0 populations of 2500 animals were mixed with 1 ml of 10× saturated *E. coli* OP50-1 or *P. vranovensis* and a low dose of *N. parisii* spores (see *Table 3*) and plated on a 10 cm plate. This low dose limited the detrimental effects on animal fertility that are observed with higher doses, while ensuring most animals were still infected. F1 populations of 1000 animals were mixed with 400 μl of 10× saturated *E. coli* OP50-1 and a high dose of *N. parisii* spores (see *Table 3*) and plated on a 6 cm plate.

To test for inherited immunity to *N. parisii* in *C. elegans*, *C. briggsae*, *C. tropicalis*, and *C. kamaaina*, synchronized animals were infected from the L1 larval stage with a low dose of *N. parisii*. *C. elegans* and *C. briggsae* were grown for 72 hr at 21°C; *C. tropicalis* and *C. kamaaina* were

**Table 3.** Details of *N. parisii* doses employed.

| *N. parisii* dose | Plate concentration (spores/cm²) | Millions of spores used | |
|---|---|---|---|
| | | **6 cm plate** | **10 cm plate** |
| Low | ~32,000 | | 2.5 |
| High | ~88,000 | 2.5 | |

grown for 96 hr at 21°C. Ten percent of total P0 animals were fixed in acetone for DY96 staining, as described below. Embryos from the remaining animals were collected by bleaching and synchronized by hatching overnight in M9. Resulting F1 animals were infected from the L1 larval stage with a high dose of *N. parisii*. *C. elegans* and *C. briggsae* were fixed at 72 hr postinfection (hpi) at 21°C; *C. tropicalis* and *C. kamaaina* were fixed at 96 hpi at 21°C.

For multiple stress adaptation assays using *N. parisii* and osmotic stress, animals were grown on NGM agar plates seeded with 10× saturated *E. coli* OP50-1 until the L4 stage. Next, animals were collected and mixed with 1 ml of either *E. coli* OP50-1 alone or supplemented with a low dose of *N. parisii* spores and plated on either 50 mM NaCl or 250 mM NaCl plates. Animals were grown for 24 hr at 21 °C. Embryos from these animals were collected by bleaching. To test adaptation to osmotic stress, 2000 F1 embryos were transferred to 420 mM NaCl plates seeded with *E. coli* OP50-1. Percentage of animals hatched was scored after 48 hr at 21 °C, as previously described in *Burton et al., 2017* and *Burton et al., 2020*. To test adaptation to *N. parisii*, the remaining embryos were synchronized by hatching overnight in M9. Resulting F1 animals were either not infected as controls, or infected at the L1 larval stage with a high dose of *N. parisii*. Animals were fixed after 72 hr at 21 °C for DY96 staining and analysis.

For multiple stress adaptation assays using *N. parisii* and *P. vranovensis*, animals were grown on NGM agar plates seeded with *E. coli* OP50-1 until the L4/young adult stage. Next, animals were collected and mixed with 1 ml of either *E. coli* OP50-1 alone or *E. coli* OP50-1 supplemented with a low dose of *N. parisii* spores, or 1 ml of *P. vranovensis* BIGb0446 alone or *P. vranovensis* BIGb0446 supplemented with a low dose of *N. parisii* spores. Animals were plated on NGM and grown for 24 hr at 21 °C. Embryos from these animals were collected by bleaching. To test adaptation to *P. vranovensis*, 2000 F1 embryos were transferred to new NGM plates seeded with *P. vranovensis* BIGb0446. Percentage of animals surviving was scored after 24 hr at 21 °C as previously described in *Burton et al., 2017* and *Burton et al., 2020*. To test adaptation to *N. parisii*, the remaining embryos were synchronized by hatching overnight in M9. Resulting F1 animals were either not infected as controls, or infected from the earliest larval stage with a high dose of *N. parisii*. Animals were fixed after 72 hr at 21 °C for DY96 staining and analysis.

## Fixation and staining of *N. parisii* infection

Worms were washed off plates with M9 and fixed in 1 ml acetone for 10 min at room temperature, or overnight at 4 °C. Fixed animals were washed twice in 1 ml PBST (phosphate-buffered saline [PBS] containing 0.1 % Tween-20) before staining. Microsporidia spores were visualized with the chitin-binding dye Direct Yellow (DY96). For DY96 staining alone, animals were resuspended in 500 µl staining solution (PBST, 0.1 % sodium dodecyl sulfate, 20 µg/ml DY96), and rotated at 21 °C for 30 min in the dark. DY96-stained worms were resuspended in 20 µl EverBrite Mounting Medium (Biotium) and mounted on slides for imaging. Note: to pellet worms during fixation and staining protocols, animals were centrifuged for 30 s at 10,000 × *g*.

## Image analysis of *N. parisii* infection

Worms were imaged with an Axioimager 2 (Zeiss). DY96-stained worms were imaged to determine number of embryos per worm. Worms possessing any quantity of intracellular DY96-stained microsporidia were considered infected. Precise microsporidia burdens were determined using ImageJ/FIJI (*Schindelin et al., 2012*). For this, each worm was defined as an individual 'region of interest' and fluorescence from GFP (DY96-stained microsporidia) subject to 'threshold' and 'measure area percentage' functions on ImageJ. Images were thresholded to capture the brighter signal from microsporidia spores, while eliminating the dimmer GFP signal from worm embryos. Final values are given as % fluorescence for single animals.

## Preparation of OP50 for plating worms

One colony of *E. coli* strain OP50 was added to 100 ml of LB and grown overnight at room temperature then stored at 4 °C. One or five drops of HB101 were added to 6 or 10 cm plates of NGM, respectively, to use for growing worm strains and recovering them from starvation.

## Preparation of HB101 for liquid culture

One colony of *E. coli* strain HB101 was added to a 5 ml starter culture of LB with streptomycin and grown for 24 hr at 37 °C. The starter cultures were then added to a 1 l culture of TB and grown for another 24 hr at 37 °C. The bacteria was centrifuged for 10 min at 5000 rpm to form a pellet. After being weighed, the bacteria was then resuspended in S-complete to create a 10× (250 mg/ml) stock that was stored at 4 °C. Further dilutions with S-complete were used to create the dilutions for each condition in this experiment.

## Dietary restriction/dilution series cultures

For *C. elegans*, *C. briggsae*, and *C. tropicalis*, 10 L4 hermaphrodite worms were picked onto three 10 cm plates seeded with OP50, and for *C. kamaaina* 10 L4 females and ~20 males were picked onto three 10 cm plates. For all species, adults were removed after 24 hr. *C. elegans* and *C. briggsae* were grown for 96 hr before bleaching and *C. tropicalis* and *C. kamaaina* were grown for 120 hr before bleaching due to slower growth and longer generation time. After bleaching, worms were aliquoted into 100 ml cultures of S-complete at one worm/100 µl with a concentration of 25 , 12.5 , 6.25 , 3.13 , or 1.6 mg/ml of HB101 and kept in 500 ml flasks in shaking incubators at 20 °C and 180 rpm. Worms were grown in these cultures for 96 hr (*C. elegans*), 102 hr (*C. briggsae*), or 120 hr (*C. tropicalis* and *C. kamaaina*) before being bleached and prepared for starvation cultures. Due to slow development and inability to properly scale up in liquid culture, 1.6 mg/ml cultures for *C. briggsae* and 1.6 and 3.13 mg/ml cultures for *C. kamaaina* were excluded from the rest of this experiment.

## Starvation cultures

After bleach, embryos were placed into 5 ml virgin S-basal cultures in 16 mm glass test tubes on a roller drum at 20 °C at one worm/µl. Worms were aliquoted out of this culture using micropipettes for further assays.

## Measuring L1 size

Twenty-four hours after bleach (~12 hr after hatch), 1000 L1s were pipetted out of the starvation cultures, spun down in 15 ml plastic conical tubes by centrifuge for 1 min at 3000 rpm then plated onto unseeded 10 cm NGM plates. L1s were imaged with a Zeiss Discovery. V20 stereomicroscope at ×77 and measured using Wormsizer (*Moore et al., 2013*). Ad libitum concentration was defined as 25 mg/ml and dietary restriction concentration was determined based on what concentration of HB101 produced the largest average L1 size for each strain. For *C. elegans*, this was 3.13 mg/ml, and eightfold dilution from ad libitum and consistent with previous determinations for dietary restriction in *C. elegans* (*Hibshman et al., 2016*). For *C. briggsae*, peak L1 size varied between 12.5 and 6.25 mg/ml depending on replicate. We chose to use 6.25 mg/ml as the dietary restriction concentration to be consistent with replicates that were already being processed. The peak L1 size and determination of dietary restriction for *C. tropicalis* were 6.13 mg/ml. *C. kamaaina* did not show a significant change in L1 size across conditions and was ultimately excluded from the brood size assay due to difficulty interpreting effects of starvation on brood size in a male–female strain.

## L1 size statistics

A linear mixed effects model was performed on the L1 size data to see if there was a significant effect of HB101 concentration on L1 size. The lme4 package in R studio was used to perform this linear mixed effects test. The function lmer() was used on data from each species, for example: • lmer(length~ condition + (1 | replicate) + (1 | replicate:condition), data = C_elegans), 'length' is the length in microns of each individual worm, 'condition' is the fixed effect of the concentration of HB101, '1 | replicate' is the addition of the random effect of replicate to the model, '1 | replicate:condition' is the addition of the random effect per combination of replicate and condition, and 'data' is the primary spreadsheet restricted by the species of interest.

## Gene orthology inference among species

To identify one-to-one orthologs across the four species, we downloaded protein and GFF3 files for *C. elegans*, *C. briggsae*, and *C. tropicalis* genomes from WormBase (*Harris et al., 2020*) (version WS275) and for the *C. kamaaina* genome from *caenorhabditis*.org (version v1). We assessed gene

set completeness using BUSCO (*Simão et al., 2015*) (version 4.0.6; using the parameter *-m proteins*) using the 'nematoda_odb10' lineage dataset. For each species, we selected the longest isoform for each protein-coding gene using the agat_sp_keep_longest_isoform.pl script from AGAT (*Jacques Dainat, 2021*) (version 0.4.0). Filtered protein files were clustered into orthologous groups (OGs) using OrthoFinder (*Emms and Kelly, 2019*) (version 2.4.0; using the parameter *-og*) and one-to-one OGs were selected.

## F1 and F3 sample collection for RNA-seq

Young adult animals grown on NGM agar plates seeded with *E. coli* HB101 were collected and transferred to new plates seeded with either control plates (50 mM NaCl) seeded with *E. coli* HB101, *P. vranovensis* BIGb0446, *P. vranovensis* BIGb0427, *S. plymuthica* BUR1537, *Pseudomonas* sp. 15C5, *Aeromonas* sp. BIGb0469, or plates containing 300 mM NaCl seeded with *E. coli* HB101. Animals were grown for 24 hr at room temperature (22 °C). Embryos from these animals were collected by bleaching and immediately frozen in 1 ml Trizol.

## Analysis of RNA-seq data

RNA libraries were prepared and sequenced by BGI TECH SOLUTIONS using 100PE DNBseq Eukaryotic Transcriptome service. Quality controlled and adapter trimming of RNA reads were performed using fastp-v4.20.0 (*Chen et al., 2018*) (`--qualified_quality_phred 20 --unqualified_percent_limit 40 --length_required 50 --low_complexity_filter --complexity_threshold 30 --detect_adapter_for_pe --correction --trim_poly_g --trim_poly_x \ --trim_front1 2 --trim_tail1 2 --trim_front2 2 --trim_tail2 2`) (1). Next, reads were aligned using STAR-2.7.1a (*Dobin et al., 2013*) (`--alignSJoverhangMin 8 --alignSJDBoverhangMin 1 --outFilterMismatchNmax 999 --outFilterMismatchNoverReadLmax 0.04 --alignIntronMin 10 --alignIntronMax 1000000 --alignMatesGapMax 1000000 --outFilterType BySJout --outFilterMultimapNmax 10000 --winAnchorMultimapNmax 50 --outMultimapperOrder Random`) (2) against the genome of *C. elegans* WS275, *C. briggsae* WS275, *C. tropicalis* WS275, and the *C. kamaaina* genome obtained from caenorhabditis.org. Read counts were obtained using subread-2.0.0 (-M -O -p `--fraction -F GTF -a -t exon -g gene_id`) (*Liao et al., 2014*) (3) using the annotation for *C. elegans* PRJNA13758.WS275, *C. briggsae* PRJNA10731.WS275, *C. tropicalis* PRJNA53597.WS275, and *C. kamaaina* Caenorhabditis_kamaaina_QG2077_v1. Counts were imported into R and differential gene expression analysis was performed with DESeq2 (FDR < 0.01) (*Love et al., 2014*).

For comparisons made between different species, genes were subsetted to include only those 7587 single-copy ortholog groups that were identified between the four species. In addition to the 7203 genes that were identified as single-copy ortholog groups by OrthoFinder, the 7587 contain an additional 385 ortholog groups that were identified as having more than one ortholog in one out four of the species but where all but one of the multiple orthologs had no observable expression in any of the samples collected.

For the comparison between the stress response and gene expression during embryo development, data were downloaded from *Boeck et al., 2016* and imported in R with raw counts from this study. The range of embryo expression for each gene was considered as one standard deviation ± the mean of regularized log normalized counts across all embryo time points. DEGs from the stress experiments where the regularized log normalized counts for one or both of the comparison samples (for all replicates) were outside of the embryo range were considered unlikely to be caused by developmental timing.

## L4+ developmental rate assays

Young adult animals that were grown on NGM agar plates seeded with *E. coli* HB101 were collected and transferred to new plates seeded with *E. coli* HB101, *Pseudomonas* sp. 15C5, *S. plymuthica* BUR1537, or *Aeromonas* sp. BIGb0469. Animals were grown for 24 hr at room temperature (22 °C). Embryos from these animals were collected by bleaching and transferred to new plates seeded with 1 ml of *E. coli* HB101 *Pseudomonas* sp. 15C5, *S. plymuthica* BUR1537, or *Aeromonas* sp. BIGb0469. Percent of animals that reached the L4 larval stage was scored after either 48 or 72 hr at 22 °C.

## Identification of *Pseudomonas* sp. 15 C5 and *S. plymuthica* BUR1537

Samples of rotting fruit and vegetation were collected from around Cambridge (UK) in 50 ml vials. For isolation of wild bacteria, the samples were homogenized and resuspended in M9 and plated on LB Agar, Nutrient Agar, or Actinomycete Isolation Agar plates and grown at either 37 °C or 30 °C for 24 hr. Single colonies were isolated from the plates and grown in LB or Nutrient Broth at the same temperature overnight. Stocks were frozen and stored at −80 °C in 20 % glycerol. One thousand five hundred and thirty-seven total isolates were obtained and frozen. *C. elegans* embryos were placed onto NGM agar plates seeded with each of the 1537 bacterial isolates. Bacterial isolates that caused substantial delays in animal development or lethality were further analyzed for isolates where parental exposure to the isolate for 24 hr modified offspring phenotype when compared to offspring from parents fed the normal laboratory diet of *E. coli* HB101. Bacterial genus and species were identified by 16 S rRNA profiling and sequencing.

## RNAi in *C. kamaaina*

dsDNA corresponding to the *C. kamaaina* orthologs of *cysl-1*, *rhy-1*, *mek-2*, and *gpdh-2* was synthesized and cloned into the L4440 vector by GENEWIZ (Takeley, UK). Vectors were transformed in *E. coli* HT115. *C. kamaaina* embryos were collected by bleaching and placed onto NGM agar plates containing 1 mM IPTG that were seeded with *E. coli* HT115 transformed with either the L4440 empty vector or each of the new vectors and grown at room temperature (22 °C) for 48 hr. After 48 hr, animals were transferred to new 50 mM NaCl control plates seeded with *E. coli* HB101, 300 mM NaCl plates seeded with *E. coli* HB101, or 50 mM NaCl control plates seeded with *P. vranovensis* BIGb0446. Animals were grown for 24 hr at room temperature (22 °C). Embryos from these animals were collected by bleaching and transferred to new 500 mM NaCl plates seeded with *E. coli* HB101 or 50 mM NaCl plates seeded with *P. vranovensis* BIGb0446. Percent of animals developing or surviving was scored after 24 hr at room temperature as previously described in *Burton et al., 2017* and *Burton et al., 2020*.

## Statistics and reproducibility

Sample sizes for experiments involving *C. elegans* were selected based on similar studies from the literature and all animals from each genotype and condition were grouped and analyzed randomly. All replicate numbers listed in figure legends represent biological replicates of independent animals cultured separately, collected separately, and analyzed separately. Unpaired two-tailed Student's *t*-test was used for *Figures 1B, D, F, 2E, J, 4C and D*, and *Figure 1—figure supplement 1F-G*. Two-way ANOVA was used for *Figures 1C, E and 3A–G*, and *Figure 1—figure supplement 1A-E*. One-way ANOVA was used for *Figure 4A and B*, and *Figure 4—figure supplement 1*. *$p < 0.05$, **$p < 0.01$, ***$p < 0.001$, ****$p < 0.0001$. The experiments were not randomized. The investigators were not blinded to allocation during experiments and outcome assessment.

## Acknowledgements

We would like to thank Buck Samuel and Marie-Anne Felix for bacterial isolates. We also thank Matt Rockman and Luke Noble for prepublication access to the genome of *C. kamaaina*. We would also like to thank Marie-Anne Felix and the *Caenorhabditis* Genetic Center, which is funded by the NIH National Center for Research Resources (NCRR), for *Caenorhabditis* strains. NOB is funded by a Next Generation Fellowship from the Centre for Trophoblast Research. KF and LRB were funded by the National Institutes of Health (GM117408, LRB). AW and AR were funded by the Natural Sciences and Engineering Research Council of Canada (Grant #522691522691) and an Alfred P Sloan Research Fellowship FG2019-12040 (to AWR). This work was also supported by Cancer Research UK (C13474/A18583, C6946/A14492) and the Wellcome Trust (104640/Z/14/Z, 092096/Z/10/Z) grants to EAM.

## Additional information

### Funding

| Funder | Grant reference number | Author |
|---|---|---|
| Centre Trophoblast Research | Next Generation fellowship | Nicholas O Burton |
| National Institutes of Health | | Kinsey Fisher<br>L Ryan Baugh |
| National Institutes of Health | GM117408 | L Ryan Baugh |
| Natural Sciences and Engineering Research Council of Canada | Grant #522691522691 | Alexandra Willis<br>Aaron W Reinke |
| Alfred P. Sloan Foundation | FG2019-12040 | Aaron Reinke |
| Cancer Research UK | C13474/A18583 | Eric A Miska |
| Cancer Research UK | C6946/A14492 | Eric A Miska |
| Wellcome Trust | 104640/Z/14/Z | Eric A Miska |
| Wellcome Trust | 092096/Z/10/Z | Eric A Miska |

The funders had no role in study design, data collection and interpretation, or the decision to submit the work for publication.

### Author contributions

Nicholas O Burton, Alexandra Willis, Conceptualization, Data curation, Formal analysis, Funding acquisition, Investigation, Methodology, Project administration, Supervision, Writing – original draft, Writing – review and editing; Kinsey Fisher, Data curation, Formal analysis, Methodology, Writing – review and editing; Fabian Braukmann, Jonathan Price, Lewis Stevens, Data curation, Formal analysis, Investigation, Methodology, Writing – review and editing; L Ryan Baugh, Eric A Miska, Formal analysis, Funding acquisition, Methodology, Project administration, Writing – review and editing; Aaron Reinke, Data curation, Formal analysis, Funding acquisition, Investigation, Project administration, Writing – review and editing

### Author ORCIDs

Nicholas O Burton http://orcid.org/0000-0002-5495-3988
Jonathan Price http://orcid.org/0000-0001-6554-5667
Lewis Stevens http://orcid.org/0000-0002-6075-8273
L Ryan Baugh http://orcid.org/0000-0003-2148-5492
Aaron Reinke http://orcid.org/0000-0001-7612-5342
Eric A Miska http://orcid.org/0000-0002-4450-576X

### Decision letter and Author response

Decision letter https://doi.org/10.7554/eLife.73425.sa1
Author response https://doi.org/10.7554/eLife.73425.sa2

## Additional files

### Supplementary files

• Supplementary file 1. List of 7587 single-copy orthologous genes conserved among *C. elegans*, *C. briggsae*, *C. kamaaina*, and *C. tropicalis*.

• Supplementary file 2. Expression of single-copy orthologous genes in F1 progeny of animals exposed to *P. vranovensis*.

• Supplementary file 3. Expression of single-copy orthologous genes in F1 progeny of animals exposed to osmotic stress.

• Supplementary file 4. Expression of single-copy orthologous genes in F3 progeny of animals

exposed to *P. vranovensis* and osmotic stress.

- Supplementary file 5. List of bacteria isolated from United Kingdom.
- Supplementary file 6. PCR sequences of *Pseudomonas* 15C5 16 S rRNA and *rpoD*.
- Supplementary file 7. Expression of single-copy orthologous genes in F1 progeny of *C. elegans* exposed to *P. vranovensis*, *Pseudomonas* sp. 15C5, *Serratia plymuthica BUR1537*, or *Aeromonas* sp. BIGb0469.
- Transparent reporting form

### Data availability

RNA-seq data that support the findings of this study have been deposited at NCBI GEO and are available under the accession code GSE173987.

The following dataset was generated:

| Author(s) | Year | Dataset title | Dataset URL | Database and Identifier |
|---|---|---|---|---|
| Burton N, Price J, Braukmann F, Miska E | 2021 | Parental exposure to environmental stress results in evolutionarily conserved intergenerational changes in offspring gene expression | https://www.ncbi.nlm.nih.gov/geo/query/acc.cgi?acc=GSE173987 | NCBI Gene Expression Omnibus, GSE173987 |

The following previously published datasets were used:

| Author(s) | Year | Dataset title | Dataset URL | Database and Identifier |
|---|---|---|---|---|
| Boeck M | 2016 | The time-resolved transcriptome of C. elegans | https://www.ncbi.nlm.nih.gov/pmc/articles/PMC5052054/ | NCBI Sequence Read Archive - Supplemental Table 1, PMC5052054 |

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
