## [Decision Letter]

**Acceptance summary:**

The authors have systematically studied the question of an evolutionary conservation of stress responses in different species of nematodes. They identify some common and stress-specific pathways that lead to modifications in offspring of stressed parents. The study provides much needed experimental data on the question of how such epigenetic changes could contribute to evolutionary adaptive processes.

**Decision letter after peer review:**

[Editors’ note: the authors submitted for reconsideration following the decision after peer review. What follows is the decision letter after the first round of review.]

Thank you for submitting the paper "Intergenerational adaptations to stress are evolutionarily conserved, stress specific, and have deleterious trade-offs" for consideration by *eLife*. Your article has been reviewed by 3 peer reviewers, and the evaluation has been overseen by a Reviewing Editor and a Senior Editor. The following individual involved in review of your submission has agreed to reveal their identity: Alexei Maklakov (Reviewer #2).

We are sorry to say that, after consultation with the reviewers, we have decided that this work will not be considered further for publication by *eLife* at this stage.

As you will see from the reviews below, all three reviewers acknowledge the great potential of your approach. However, the transcriptome data that are the basis for the bulk of the conclusions are not fully adequate and too superficially analysed. It would seem very important to have at least the parental expression data as a reference and address the problem that only a subset of the embryos shows a response (see comment of reviewer 1). Further, with only three biological replicates, it would be necessary to use a more stringent cut-off for calling significant genes, or to use independent confirmation experiments. A more explicit pathway analysis should be also included. Also, the discussion (and possibly further analysis) of genes showing transgenerational effects should be enhanced. Since it is not clear whether the additional RNASeq data can be generated in a short time, we decided to reject at this point, but we would be happy to reconsider a significantly updated version.

*Reviewer #1:*

The general idea of comparing response patterns to stress in the offspring generation is new and very interesting. However, the data that are presented are in several ways preliminary. The phenotype comparisons are mostly convincing, although statistical treatments are partly unclear, given that each "replicate" includes itself many individuals. The transcriptomic data are minimal (only three replicates) and lack comparison to the stress responses in the parental animals. The analysis of the transcriptome data is limited to counting overlaps between significantly changed genes, without deeper discussion of the genes and pathways that are affected. The top response genes that are directly tested have been discovered before. Hence, while interesting patterns are evident from the data, this work largely confirms prior work, including that described in Burton et al. 2020.

The title and the discussion claim evolutionary conservation, but the actual results support this only partly. This needs to be stated more carefully. Also, given that the species used are rather closely related, one should relate the level of "conservation" to other conserved processes between these species.

While the term "intergenerational" is increasingly often used in these studies, the authors focus essentially on classic parental effects. A more systematic comparison with further generations would be useful to judge whether epigenetic programming might occur, or whether it is a pure F1 effect. In fact, given that there is at least a small subset of genes that show transgenerational effects, one should have expected a much deeper analysis of these genes.

The RNASeq data are produced on batches of embryos, on which only a subset (e.g. 50% for the infections) would show an effect. It would be very important to obtain single-embryo data to better understand why some react and some not.

The analysis of the RNASeq data is too superficial. Given that there are only three biological replicates, in a situation where much variance is to be expected (see comment above), one has to consider the data as highly underpowered. Hence, it would be necessary to include not only a p-value cut-off, but also a "fold-expression change" cut-off (e.g. at least 2-fold or even more). This would lead to rather different numbers and possibly also to different conclusions.

*Reviewer #2:*

Transgenerational effects (TE) (usually defined as multigenerational effects lasting for at least three generations) generated a lot of interest in recent years but the adaptive value of such effects is unclear. In order to understand the scope for adaptive TE we need to understand (i) whether such effects are common; (ii) whether they are stress-specific; and (iii) if there are trade-offs with respect to performance in different environments. The last point is particularly important because F1, F2 and F3 descendants may encounter very different environments. On the other hand, intergenerational effects (lasting for one or two generations) are relatively common and can play an important role in evolutionary processes. However, we do not know whether intergenerational and transgenerational effects have same underlying mechanisms.

This study makes a big step towards resolving these questions and strongly advances our understanding of both phenomena. Much of the previous work on mechanisms of multigenerational effects has been conducted in *C. elegans* and this works uses the same approach. They focus on bacterial infection, Microsporidia infection, larval starvation and osmotic stress. I did not quite understand why the authors chose to focus on P. vranovensis rather than *P. aeruginosa* P14 that has been used in previous studies of transgenerational effects in *C. elegans*. However, this is a minor point because I guess they were interested in broad transgenerational responses to bacterial infection rather than in strain-specific ones. The authors used different Caenorhabditis species, which is another strength of this study in addition to using multiple stresses.

They found 279 genes that exhibited intergenerational changes in all C species tested, but most interestingly, they show that a reversal in gene expression corresponds to a reversal in response to bacterial infection (beneficial in two species and deleterious on one). This is very intriguing! This was further supported by similar observations of osmotic stress response.

They also report that intergenerational effects are stress-specific and there have deleterious effects in mismatched environments, and, importantly, when worms were subject to multiple stresses. It is quite likely that offspring will experience a range of environments and that several environmental stresses will be present simultaneously in nature. I really liked this aspect of this work as I think that tests in different environments, especially environments with multiple stresses, are often lacking, which limits the generality of the conclusions.

Another interesting piece of the puzzle is that beneficial and deleterious effects could be mediated by the same mechanisms. It would be interesting to explore this further. However, this is not a real criticism of this work. I think that the authors collected an impressive dataset already and every good study generates new research questions.

Given these findings, I was particularly keen to see what comes of transgenerational effects. The general answer was that there aren't many, and the authors conclude that all intergenerational effects that they studied are largely reversible and that intergenerational and transgenerational effects represent distinct phenomena. While I think that this is a very important finding, I am not sure whether we can conclude that intergenerational and transgenerational effects are not related.

In my view, an alternative interpretation is that intergenerational effects are common while transgenerational effects are rare. Because intergenerational effects are stress-specific, transgenerational effects could be stress-specific as well.

Perhaps different mechanisms regulate intergenerational responses to, say, different forms of starvation (e.g. compare opposing transgenerational responses to prolonged larval starvation (Rechavi et al. doi:10.1016/j.cell.2014.06.020) and rather short adulthood starvation (Ivimey-Cook et al. 2021 https://doi.org/10.1098/rspb.2021.0701)). Perhaps some (most?) forms of starvation generate only intergenerational responses and do not generate transgenerational responses. But some do. Those forms of starvation that generate both intergenerational and transgenerational effects could do so via same mechanisms and represent the same phenomenon. I am by no means saying this is the case, but I am not sure that the absence of evidence of transgenerational effects in this study necessarily suggests that inter- and trans-generational effects are different phenomena.

The only concern real concern was the lack of phenotypic data on F3 beyond gene expression. Ideally, I would like to see tests of pathogen avoidance and starvation resistance in F3. However, given the amount of work that went into this study, the lack of strong signature of potential transgenerational effects in gene expression, and the fact that most of these effects were shown previously to last only one generation, I do not think this is crucial.

It would be very interesting to compare gene expression and other phenotypic responses in F1 and F3 between P. vranovensis and PA14. Also, it would be interesting to test the adaptive value of intergenerational and transgenerational effects after exposure to both strains in different environments. This is would be very informative and help with understanding the evolutionary significance of transgenerational epigenetic inheritance of pathogen avoidance as reported previously. Why response to P. vranovensis is erased while response to PA14 is maintained for four generations? Are nematodes more likely to encounter one species than the other? Again, however, this is not something necessary for this study.

The main strengths of this paper are (i) use of multiple stresses; (ii) use of multiple species; (iii) tests in different environments; and (iv) simultaneous evaluation of intergenerational and transgenerational responses. This study is first of a kind, and it provides several important answers, while highlighting clear paths for future work.

Excellent work and I think it will generate a lot of interest in the community, definitely want to see it published in *eLife*.

I am not fully convinced about their interpretation of whether inter- transgenerational effects are separate phenomena but happy to be convinced otherwise.

*Reviewer #3:*

In this manuscript, the authors address whether the mechanisms mediating intergenerational effects are conserved in evolution. This question is important not only to frame this phenomenon in an evolutionary context, but to address several interlinked questions: is there a mechanism in common between adaptive versus deleterious effects? What makes some effects last one instead of several generations? What is the ecological relevance for those mechanisms? Using *Caenorhabditis elegans* as a model of reference, they compare four types of intergenerational effects on additional three Caenorhabditis species.

The authors used previously characterized models of intergenerational inheritance, focusing on those that are likely to have adaptive significance. This is relevant, because the adaptive relevance of other published examples of inter- and transgenerational inheritance is not clear. They used functional studies to probe for conservation of mechanisms for bacterial infection and resistance to osmolarity stress, which is a major strength of this study. The data supports the claim of conservation in some types of intergenerational inheritance and divergence in others. One major question addressed in this manuscript is whether there is a potential overarching mechanism that confers stress-resistance across generations. Their experiments convincingly show that this is not the case, but that instead, there are stress-specific mechanisms responsible for intergenerational inheritance.

The authors highlighted the discovery of 279 highly conserved genes that exhibited intergenerational in gene expression. The manuscript would be strengthened if these genes were shown to have functional relevance.

It is interesting that *C. elegans* can pass the P. vavronensis resistance to the offspring, but not C. briggsae. Is this because C. briggsae is already expressing cysl-1 in the parental generation? What would happen if cysl-1 would be knocked out in this species? Would it lack resistance to the pathogen?

The authors showed that *C. elegans* and C. briggsae seem to share intergenerational effects of resistance to N. parisii in other species. It would be interesting to know if the mechanisms are conserved, but unfortunately this information is lacking. Similarly, it would be interesting to know if gdph-2 mediates resistance to osmolarity stress in C. kamaaina.

The supplementary tables would be more useful if the three-letter name would be included as well.

Indicate in Figure 3C that these are RNAi experiments. Also for Figure 3C, spell out "EV" in the legend of the figure. I suppose it means 'empty vector'?

The font size in the figures is pretty small and therefore difficult to read.

---

## [Author Response]

[Editors’ note: the authors resubmitted a revised version of the paper for consideration. What follows is the authors’ response to the first round of review.]

Reviewer #1:The general idea of comparing response patterns to stress in the offspring generation is new and very interesting.

We thank Reviewer 1 for their time and thoughtful comments. We agree that these comparisons are new and very interesting and have added multiple revised analyses to the manuscript based on the reviewer comments that we think will further enhance the impact of and conclusions made in this study.

However, the data that are presented are in several ways preliminary. The phenotype comparisons are mostly convincing, although statistical treatments are partly unclear, given that each "replicate" includes itself many individuals.

The statistical treatments for groups of individuals are the same as in Burton et al., 2017, Burton et al., 2020, and Willis et al., 2021 which include the original reports of the intergenerational responses studied here. Replicates that include many individuals are relatively common when working with *C. elegans* and are usually compared using ANOVA or student’s t-tests (depending on the number of comparisons) to analyze the variation in batch effects as well as differences between populations of animals.

We believe this ability to assay hundreds or even thousands of animals, in total, for each comparison in this study makes our data substantially stronger and more reliable. However we are happy to perform any additional statistical tests the reviewer might want to see.

The transcriptomic data are minimal (only three replicates)

To address this comment we compared our original three replicates of RNA-seq from F1 animals from *C. elegans* parents exposed to *P. vranovensis* BIGb0446 to a second independent three replicates of F1 animals from *C. elegans* parents exposed to a second *P. vranovensis* isolate (BIGb0427 – the data for this second P. vranovensis isolate was already part of Figure 4 of this manuscript).

By comparing these three new replicates to our previous findings from three original replicates we found that 515 of the 562 genes that exhibited a >2-fold change and were significant at padj <0.01 in the original three replicates were also changed at >2-fold and padj <0.01 in the new three replicates. We believe our findings that 91.6% of genes change >2-fold and remain significant at padj<0.01 even when the number of replicates is doubled (and a different isolate of *P. vranovensis* is used!) suggests that adding additional replicates would not substantially change the conclusions of this manuscript.

We would also like to highlight, as above, that because this analysis was done on populations of thousands of similarly staged animals, as opposed to individuals, that this further reduces the variability between replicates. In addition, much of our transcriptomic data from each species was then compared across species and genes were only analyzed for those that changed in multiple different species which themselves each represent a separate three additional replicates [ie genes that change in all 4 species analyzed have to exhibit significant (>2-fold, padj <0.01) changes across 12 total replicates].

Our new findings comparing six replicates did not substantially change the number of genes identified when compared to using three replicates, and the fact that for all of the main conclusions of this manuscript each set of triplicates from one species was then compared across 9 additional replicates from three other species from pools of thousands of animals makes us very confident that our results are robust and highly reproducible.

… and lack comparison to the stress responses in the parental animals.

We agree with Reviewer 1 that comparisons to parental animals are interesting and important. Comparisons of F1 progeny gene expression patterns to parental animals were not included here because such comparisons were previously published in some of our original reports of these intergenerational effects (For example, see Burton et al., 2020). In summary, we found that most, but not all, of the effects on gene expression in F1 animals were also detected in parental animals. However, the transcriptional responses only turn on in F1 animals post gastrulation and do not appear to be due to the simple deposition of parental mRNAs into embryos (Burton et al., 2020).

We have updated the text to highlight these findings.

The analysis of the transcriptome data is limited to counting overlaps between significantly changed genes, without deeper discussion of the genes and pathways that are affected.

In the revised manuscript we have completely redone all of the transcriptomic analysis to use a stricter set of cutoffs for significance – both padj <0.01 and requiring a >2-fold change in expression based on the helpful comments of Reviewer 1 – which we agree with – see below.

As part of this new analysis we have now also included a deeper discussion of the genes that exhibited similar changes across species, including using g:Profiler to examine the genes that exhibited changes across all four species.

In addition, we have now paired our phenotypic and transcriptomic data across species to identify 19 new genes that we predict are highly likely to be involved in intergenerational responses to stress based on their expression patterns across species. These 19 genes come out of highly filtered analyses across species that identified a total of 23 genes that change only in species that adapt to P. vranovensis or osmotic stress and not in species that do not adapt.

Interestingly, this analysis identified nearly all of the previously known genes involved in intergenerational adaptations to these stresses including rhy-1, cysl-1, cysl-2 and gpdh-1. Thus, we predict the remaining 19 genes that came out of this analysis are highly likely to be involved in the responses to these stresses. Furthermore, in the revised text we highlight that our new list of 19 genes includes multiple conserved factors that are required for animal viability including genes involved in nuclear transport (imb-1 and xpo-2), the CDC25 phosphatase ortholog cdc-25.1, and the PTEN tumor suppressor ortholog daf-18. This new analysis will likely form the basis for future investigations into the mechanisms underlying these exciting intergenerational effects.

We believe this additional analysis greatly improves this manuscript. We are also happy to include any specific additional analysis the reviewer would like to see.

The top response genes that are directly tested have been discovered before. Hence, while interesting patterns are evident from the data, this work largely confirms prior work, including that described in Burton et al. 2020.

We have revised the text to highlight that the aims of this particular study were to determine if multigenerational responses to stress were evolutionarily conserved at any level, as well as to determine the potential costs of such effects and the specificity of the responses. Questions that were not addressed in any previous study of multigenerational effects, including Burton et al., 2020.

Because of the aims of this study we believe it was critical to focus on genes that had an established role in these intergenerational responses in *C. elegans* and to compare and contrast the behavior and requirement of these genes in intergenerational responses in other species. (Although we note that this newly revised manuscript, we have now also reported 19 new top response genes – see above).

In addition to our original goals, in this study we were able to determine the extent to which intergenerational transcriptional responses are conserved and the extent to which intergenerational transcriptional changes persist trans-generationally (which we find to be effectively not at all using our revised stricter analysis). We believe these findings are not only novel, but perhaps will be surprising to much of the intergenerational and transgenerational field and have a major impact on both how multigenerational studies are interpreted and how they are conducted in the future. This is especially the case for studies in *C. elegans* which is one of the leading model organisms to study the mechanisms underlying both intergenerational and transgenerational responses to stress.

For example, we note that several landmark studies of transgenerational effects (persisting into F3 or later generations) in *C. elegans* performed RNA-seq on F1 progeny (For example, Moore et al., Cell 2019 or Ma et al., Nature Cell Biology 2019). Our new findings reported here suggest that it is possible that none of the transcriptional effects detected in F1 animals will persist in F3 progeny. Furthermore, our studies demonstrate the importance of comparing *C. elegans* transcriptional effects to related Caenorhabditis species as we found that only a subset of the effects detected in *C. elegans* are conserved in any other Caenorhabditis species. (Such comparisons are important for determining if and to what extent observations of intergenerational and/or transgenerational effects observed in *C. elegans* represent conserved phenomena).

For all of these reasons we believe our data is highly exciting, will be of broad interest to the field, and represent novel and potentially unexpected findings that were not previously reported in any prior work including Burton et al., 2020.

The title and the discussion claim evolutionary conservation, but the actual results support this only partly. This needs to be stated more carefully. Also, given that the species used are rather closely related, one should relate the level of "conservation" to other conserved processes between these species.

As reviewer 1 highlights, the relationship between members of a single genus is potentially small, however we would like to also highlight that worms, and Caenorhabditis species in particular, evolve much more rapidly that most species. As an example for the genomic relationship between the species used in this manuscript we would like to highlight Figure 2 from Karin Kiontke's 2004 paper (https://www.pnas.org/content/101/24/9003#sec-2) showing that the divergence between *Caenorhabditis* species is exceptionally large and they estimate that *C. elegans* relationship to *C. kamaaina* is much larger than the divergence between humans and mice for example, and is somewhere closer to the relationship between humans and zebra fish which might not be considered to be closely related.

Nonetheless, we have updated the text of this manuscript to more explicitly state the level of relationship between the four species investigated here (approximately 0.05 substitutions per site at the nucleotide level and an estimated 30 million years of evolution). We have also added a comparison to other process that diverge between these four species such as mode of reproduction (*C. elegans, C. briggsae*, and *C. tropicalis* are hermaphrodites but *C. kamaaina* is an exclusively male-female species). Lastly, we have added a statement to the text to reflect reviewer 3’s summary “The data supports the claim of conservation in some types of intergenerational inheritance and divergence in others.” We believe such a statement accurately reflects the data presented in this manuscript.

Regardless of the exact relationship between these species, we also note that the decision to focus on species within the genus *Caenorhabditis* was a conscious decision as part of our study plan because, to our knowledge, there have been no documented cases of conservation of intergenerational (or transgenerational) responses to stress in any species across any distance of relationship. Furthermore, a greater evolutionary distance would have made the transcriptional comparisons of single copy orthologs across species that we performed here more difficult as there would be fewer clear single copy orthologues due to more frequent gene duplications/deletions at greater evolutionary distances.

When we began this work it was not clear that these type of phenomena, especially the many such effects that have been described in *C. elegans*, would be conserved at all and thus we believe one of the highlights of this paper is that we can now show that these intergenerational effects are indeed conserved, at minimum, within many species of *Caenorhabditis*. Furthermore, this study creates a solid foundation to investigate these types of effects in more distantly related species.

For example, our studies using only these four related species demonstrate that pairing phenotypic and transcriptomic data for the response to bacterial infection can quickly narrow down the >1000 transcriptional changes that are detectable in *C. elegans* to a list of only 19 genes that includes all three genes previously reported to be involved in this adaptation. This finding not only predicts that the remaining genes among these 19 are highly likely to be enriched for those involved in this intergenerational adaptation but also suggests that these genes might be particularly sensitive to intergenerational regulation and thus studies in more distantly related species could potentially focus on this much smaller subset of genes.

In summary, we believe that many important and novel conclusions about the relative conservation of these types of effects can be made by comparing species within a single genus.

While the term "intergenerational" is increasingly often used in these studies, the authors focus essentially on classic parental effects.

We have modified the text to state that several of these intergenerational effects can also be described as parental effects.

We chose to use the term intergenerational here not only due to its widespread use in the literature to describe F1 effects (For example, see reviews – Perez and Lehner, N*ature Cell Biology*, 2019 or Miska and Ferguson-Smith. *Science*, 2016), but also because the term intergenerational encompasses both parental effects on F1 animals and effects that persist for two generations (and affect F2 progeny). This was important to us because some of the effects studied here, such as the response to P. vranovensis, have been found to persist for two generations under certain conditions (Burton et al., 2020) and thus cannot be described as exclusively parental in nature.

In addition to these reasons, and because we agree with the general sentiment of Reviewer 1 that the historical use of many different terms to describe different multigenerational effects can lead to much confusion, we have also recently co-authored a substantial review of non-genetic inheritance that goes in depth into the history of the terms intergenerational and transgenerational and highlights that the definition of the term intergenerational has come to include, but is not restricted to, effects that were classically described as parental effects (for a more in-depth discussion of these terms and the history of their use see the dedicated section “What are intergenerational and transgenerational effects” in our new review Burton and Greer, Seminars in Cell and Developmental Biology 2021).

Regardless, we are aware that these terms have historically been used with what are sometimes different definitions (the same is true for the term transgenerational), and thus we are also happy to change any instance of the word intergenerational in the text to parental effect should it be desired by the reviewers as we agree that many of the effects described here could also be defined as parental effects and the change in terms would not impact the exciting underlying biology or the novelty/impact of our findings.

A more systematic comparison with further generations would be useful to judge whether epigenetic programming might occur, or whether it is a pure F1 effect.

Our previous studies of these effects have already performed phenotypic comparisons with further generations (F2, F3, and beyond) and found that each of these effects are intergenerational (or parental) in nature and none of these phenotypic effects persist beyond two generations (F2) in any experimental set up we have analyzed (Burton et al., 2017; 2020, Willis et al., 2021). We have updated the text to highlight these previous analyses.

With regards to “epigenetic programming” vs. “pure F1 effect” – we do not believe these terms/effects are mutually exclusive and we do not believe the duration an effect persists for can be used to imply a specific underlying mechanism. For example, RNAi inheritance in *C. elegans*, one of the classically cited mechanisms mediating epigenetic inheritance in *C. elegans*, only persists into F1 animals and not further when the RNAi targets somatically expressed genes. (Burton et al., 2011) –[RNAi inheritance beyond F1 animals has predominantly only been observed in germ cells when silencing germline expressed genes in *C. elegans* (Buckley et al., 2012)]. Similarly, in *Arabidopsis*, intergenerational (F1 only) adaptations to stress appear to be dependent on histone modifications at stress response genes that are heritable/programmed (For example, Luna et al., 2012). We believe these effects, and others, appear to represent examples of epigenetic programming despite being pure F1 effects. (Many similar effects are reviewed in Burton and Greer, Seminars in Cell and Developmental Biology, 2021).

Along these same lines, effects that persist for more than one generation/transgenerational effects are not always clearly due to epigenetic programming. For example, there are observations of transgenerational effects in mice that are now thought to be due to the repeated transfer of microbiomes from parents to offspring (See– Sonnenburg et al., *Nature*, 2016 or the review by Perez and Lehner, *Nature Cell Biology* 2019).

While there is perhaps some debate as to what exactly qualifies as “epigenetic”, there are a growing number of examples of F1 effects of stress that are transmitted via germ cells and/or appear to represent programmed changes in gene expression that only last for a single generation and then are erased/lost after one generation across many different species. And some of these effects have been reported to require similar molecular machinery as those that persist for more than one generation.

Thus, while F1 effects could be mediated by mechanisms that are not classically considered epigenetic, we believe it is important to not rule out the possibility that such effects might be epigenetic in nature.

In fact, given that there is at least a small subset of genes that show transgenerational effects, one should have expected a much deeper analysis of these genes.

In this revised manuscript we have updated our transcriptomic analysis to be more strict on what qualifies for significance in line with Reviewer 1s comment below which we completely agree with – see below, we now required a >2-fold change in addition to a padj <0.01.

Using this revised analysis we found that none of the >1,500 changes in gene expression that were observed in F1 progeny in *C. elegans* persisted into the F3 progeny (includes both the response to *P. vranovensis* and osmotic stress). Thus our revised analysis eliminated the few transgenerational effects that did occur when the >2-fold filter was not applied and thus no deeper analysis was done.

We have, however, updated the discussion to highlight that our analysis further suggests that transgenerational effects might only occur under specific conditions, might be more rare than previously estimated, or might be mutually exclusive with intergenerational effects. In addition, we have now performed a deeper analysis on the genes that change in F1 progeny which we are very excited about given the significant, poorly understood, and apparently relatively common role intergenerational effects appear to play in organismal survival in stressful environments.

The RNASeq data are produced on batches of embryos, on which only a subset (e.g. 50% for the infections) would show an effect. It would be very important to obtain single-embryo data to better understand why some react and some not.

We agree with reviewer 1 that understanding why some, but not all offspring survive in our assays is a very interesting biological question.

In an effort to investigate if all of the embryos of parents exposed to both osmotic stress and P. vranovensis infection exhibit the observed intergenerational effects on gene expression or if only some embryos responded and not others, we have previously used GFP reporters of major stress response genes such as sod-5::GFP for osmotic stress and an endogenously tagged copy of cysl-2::GFP for P. vranovensis infection – see Burton et al., 2017 or Burton et al., 2020. This analysis, which included endogenously tagged genes, let us quantify relative gene expression responses at the single-embryo level.

We found that nearly 100% of F1 animals exhibit the increased expression of these genes in offspring at high levels even though only approximately 50% of these animals show survival in these assays and these effects (see imaging results in Burton et al., 2017 and Burton et al., 2020). These findings indicate that essentially all of the F1 animals appear to elicit, at least to some extent, the changes in gene expression that depend on a parent’s environment.

There are several possible reasons why the changes in gene expression in some F1 embryos are insufficient to lead to survival under the very stressful conditions analyzed here. We would be happy to add a discussion of the possible reasons for this to the manuscript if the reviewers would prefer this (there are many possibilities, and this type of inter-individual variability is arguably an entire field of biology). Alternatively, we could also highlight some excellent reviews of the broader field of inter-individual variability in stress responses that have been written. However, we have not added these in this current version as we felt such discussion might distract from the main conclusions of this manuscript.

The analysis of the RNASeq data is too superficial. Given that there are only three biological replicates, in a situation where much variance is to be expected (see comment above), one has to consider the data as highly underpowered. Hence, it would be necessary to include not only a p-value cutoff, but also a "fold-expression change" cutoff (e.g. at least 2-fold or even more). This would lead to rather different numbers and possibly also to different conclusions.

We agree and thank reviewer 1 for this comment. We have reanalyzed all of our transcriptomic data in this manuscript to include a >2-fold expression change cutoff. This has changed the exact numbers in the new version of this manuscript but has not changed any of the broader conclusions. In fact, this new analysis has perhaps strengthened our claims that using these more stringent cutoffs can be used to more confidently predict likely genes involved in intergenerational responses to stress.

Reviewer #2:Transgenerational effects (TE) (usually defined as multigenerational effects lasting for at least three generations) generated a lot of interest in recent years but the adaptive value of such effects is unclear. In order to understand the scope for adaptive TE we need to understand (i) whether such effects are common; (ii) whether they are stress-specific; and (iii) if there are trade-offs with respect to performance in different environments. The last point is particularly important because F1, F2 and F3 descendants may encounter very different environments. On the other hand, intergenerational effects (lasting for one or two generations) are relatively common and can play an important role in evolutionary processes. However, we do not know whether intergenerational and transgenerational effects have same underlying mechanisms.This study makes a big step towards resolving these questions and strongly advances our understanding of both phenomena. Much of the previous work on mechanisms of multigenerational effects has been conducted in C. elegans and this works uses the same approach. They focus on bacterial infection, Microsporidia infection, larval starvation and osmotic stress. I did not quite understand why the authors chose to focus on P. vranovensis rather than *P. aeruginosa* P14 that has been used in previous studies of transgenerational effects in C. elegans. However, this is a minor point because I guess they were interested in broad transgenerational responses to bacterial infection rather than in strain-specific ones. The authors used different Caenorhabditis species, which is another strength of this study in addition to using multiple stresses.

We thank the reviewer for these comments. We’d like to briefly highlight that P. vranovensis was also shown to elicit the same transgenerational effects as *P. aeruginosa* in the bioRxiv version of the same papers that reported transgenerational effects for *P. aeruginosa* (Kaletsky et al., 2020 – GRb0427 is an isolate of P. vranovensis).

It is not clear to us why this result was not included in the final published version of this manuscript, but we in fact used P. vranovensis for these studies in part because of this bioRxiv paper and because we failed to detect any robust intergenerational effects using *P. aeruginosa* PA14 in any of our assays – including at the RNA-seq level (unpublished).

Nonetheless, we have since confirmed with Coleen Murphy’s lab that they do find P. vranovensis elicits the same transgenerational effect on behaviour as *P. aeruginosa*. We expect that future investigations into the conditions under which P. vranovensis elicits effects that are lost/erased after 1 generation and the conditions under which effects might be maintained for more than 3 generations will be highly interesting.

They found 279 genes that exhibited intergenerational changes in all C species tested, but most interestingly, they show that a reversal in gene expression corresponds to a reversal in response to bacterial infection (beneficial in two species and deleterious on one). This is very intriguing! This was further supported by similar observations of osmotic stress response.

We thank Reviewer 2 for their excitement, and we agree that these findings were highly exciting.

They also report that intergenerational effects are stress-specific and there have deleterious effects in mismatched environments, and, importantly, when worms were subject to multiple stresses. It is quite likely that offspring will experience a range of environments and that several environmental stresses will be present simultaneously in nature. I really liked this aspect of this work as I think that tests in different environments, especially environments with multiple stresses, are often lacking, which limits the generality of the conclusions.Another interesting piece of the puzzle is that beneficial and deleterious effects could be mediated by the same mechanisms. It would be interesting to explore this further. However, this is not a real criticism of this work. I think that the authors collected an impressive dataset already and every good study generates new research questions.Given these findings, I was particularly keen to see what comes of transgenerational effects. The general answer was that there aren't many, and the authors conclude that all intergenerational effects that they studied are largely reversible and that intergenerational and transgenerational effects represent distinct phenomena. While I think that this is a very important finding, I am not sure whether we can conclude that intergenerational and transgenerational effects are not related.In my view, an alternative interpretation is that intergenerational effects are common while transgenerational effects are rare. Because intergenerational effects are stress-specific, transgenerational effects could be stress-specific as well.

We agree with reviewer 2 that our findings suggest that intergenerational effects are common and transgenerational effects are either rare in comparison or only occur under specific conditions. We have updated the text to include this interpretation.

Perhaps different mechanisms regulate intergenerational responses to, say, different forms of starvation (e.g. compare opposing transgenerational responses to prolonged larval starvation (Rechavi et al. doi:10.1016/j.cell.2014.06.020) and rather short adulthood starvation (Ivimey-Cook et al. 2021 https://doi.org/10.1098/rspb.2021.0701)). Perhaps some (most?) forms of starvation generate only intergenerational responses and do not generate transgenerational responses. But some do. Those forms of starvation that generate both intergenerational and transgenerational effects could do so via same mechanisms and represent the same phenomenon. I am by no means saying this is the case, but I am not sure that the absence of evidence of transgenerational effects in this study necessarily suggests that inter- and trans-generational effects are different phenomena.

We agree and, similar to above, have updated the text accordingly to state that it is also very possible that transgenerational effects only occur under certain conditions.

The only concern real concern was the lack of phenotypic data on F3 beyond gene expression. Ideally, I would like to see tests of pathogen avoidance and starvation resistance in F3. However, given the amount of work that went into this study, the lack of strong signature of potential transgenerational effects in gene expression, and the fact that most of these effects were shown previously to last only one generation, I do not think this is crucial.It would be very interesting to compare gene expression and other phenotypic responses in F1 and F3 between P. vranovensis and PA14. Also, it would be interesting to test the adaptive value of intergenerational and transgenerational effects after exposure to both strains in different environments. This is would be very informative and help with understanding the evolutionary significance of transgenerational epigenetic inheritance of pathogen avoidance as reported previously. Why response to P. vranovensis is erased while response to PA14 is maintained for four generations? Are nematodes more likely to encounter one species than the other? Again, however, this is not something necessary for this study.

We completely agree with Reviewer 2 and have indeed attempted these experiments both in Burton et al., 2020 and in unpublished results.

With regards to the transgenerational F3 effects, as mentioned above, P. vranovensis has been reported to elicit the same transgenerational effect as *P. aeruginosa* PA14 – at least as reported in the Kaletsky et al., 2020 bioRxiv version of the manuscript from the same studies. (GRb0427 is an isolate of P. vranovensis).

To date, however, in our laboratory we have been unable to detect any transgenerational effects for either P. vranovensis or *P. aeruginosa* infection on gene expression data from RNA-seq experiments (data from this manuscript and unpublished data).

It is not yet clear why this is the case, but we note that the RNA-seq analysis from the transgenerational PA14 studies (published in Moore et al., Cell 2019) was performed on F1 animals and thus was looking at intergenerational effects – to our knowledge no RNA-seq on F3 progeny from animals exposed to PA14 has ever been published. Thus, as it stands there is no existing F3 gene expression studies done using PA14 for us to compare our results to, but it remains possible that PA14 does not elicit specific effects on F3 gene expression when analyzed by RNA-seq.

For F1 effects we have published a gene expression comparison for *P. vranovensis* and *P. aeruginosa* F1 effects in a previous manuscript (Burton et al. 2020) and will add a mention of this to the text. Briefly, we detected very few F1 effects on gene expression when exposing adults to *P. aeruginosa* for 24 hours and parental infection by P. aeruginosa did not result in protection for offspring from *P. vranovensis* infection (Burton et al., 2020). We concluded that the intergenerational adaptation to *P. vranovensis* was not initiated by *P. aeruginosa* and was at least somewhat specific to *P. vranovensis* as well as the new species of *Pseudomonas* described in this manuscript which does cross protect.

The main strengths of this paper are (i) use of multiple stresses; (ii) use of multiple species; (iii) tests in different environments; and (iv) simultaneous evaluation of intergenerational and transgenerational responses. This study is first of a kind, and it provides several important answers, while highlighting clear paths for future work.Excellent work and I think it will generate a lot of interest in the community, definitely want to see it published in eLife.

We agree with Reviewer 2 and thank them for their kind comments.

I am not fully convinced about their interpretation of whether inter- transgenerational effects are separate phenomena but happy to be convinced otherwise.

We will update the text of the manuscript to highlight that it is also possible that transgenerational effects represent effects that are comparatively rare when compared to intergenerational effects as suggested by Reviewer 2 as we agree this is a valid and potentially even likely interpretation.

Reviewer #3:In this manuscript, the authors address whether the mechanisms mediating intergenerational effects are conserved in evolution. This question is important not only to frame this phenomenon in an evolutionary context, but to address several interlinked questions: is there a mechanism in common between adaptive versus deleterious effects? What makes some effects last one instead of several generations? What is the ecological relevance for those mechanisms? Using Caenorhabditis elegans as a model of reference, they compare four types of intergenerational effects on additional three Caenorhabditis species.The authors used previously characterized models of intergenerational inheritance, focusing on those that are likely to have adaptive significance. This is relevant, because the adaptive relevance of other published examples of inter- and transgenerational inheritance is not clear. They used functional studies to probe for conservation of mechanisms for bacterial infection and resistance to osmolarity stress, which is a major strength of this study. The data supports the claim of conservation in some types of intergenerational inheritance and divergence in others. One major question addressed in this manuscript is whether there is a potential overarching mechanism that confers stress-resistance across generations. Their experiments convincingly show that this is not the case, but that instead, there are stress-specific mechanisms responsible for intergenerational inheritance.

We agree and thank Reviewer 3 for their kind comments.

The authors highlighted the discovery of 279 highly conserved genes that exhibited intergenerational in gene expression. The manuscript would be strengthened if these genes were shown to have functional relevance.

We note that, in line with Reviewer 1s comments, the revised manuscript has added a >2-fold change filter to call significant gene expression changes. Due to this enhanced filter there are now 37 genes, instead of 279, that change in all species tested. In addition to these 37 genes, we have also identified 19 additional genes that specifically change >2-fold in species that adapt to *P. vranovensis* infection but do not change, or change in opposite directions, in species that do not adapt to this pathogen. Furthermore, we identified four genes that change in species that adapt to osmotic stress but not in those that do not adapt.

With regards to functional relevance, we found that all three genes previously identified as functionally required for the adaptation to *P. vranovensis* (cysl-1, cysl-2, and rhy-1) appear among the newly revised list of 19 genes that specifically change in species that adapt (*C. elegans* and *C. kamaaina*) and not others that exhibit a deleterious intergenerational effect (C. briggsae). Similarly, among the four genes that specifically change in species that adapt to osmotic stress (elegans, briggsae and kamaaina) is gpdh-1 which has previously been found to be functionally involved in adapting to osmotic stress. These data indicate that our newly revised lists generated by pairing comparative transcriptomics with our phenotypic analysis across species are highly enriched in functionally relevant genes (p < 1.337e-08 – hypergeometric probability).

We think testing the functional relevance of the remaining 19 genes on these lists will be very exciting but we have not included such analysis in this study for several reasons including the fact that many of these genes cannot be tested in the same way as cysl-1/cysl-2/rhy-1 because they are either required for embryonic viability and thus no viable mutants exist (imb-1, xpo-2, and cdc-25.1) or because they exhibit consistently decreased expression across all species (for example, daf-18) which is unlike cysl-1/cysl-2/rhy-1 which exhibit increased expression and thus could behave in phenotypically distinct ways rather than simply being required for the adaptation.

In addition, we felt such analysis was not required for, and might potentially distract from, the main aims of this study to focus on the study of the evolutionary conservation, tradeoffs, and specificity of intergenerational responses.

It is interesting that C. elegans can pass the P. vavronensis resistance to the offspring, but not C. briggsae. Is this because C. briggsae is already expressing cysl-1 in the parental generation? What would happen if cysl-1 would be knocked out in this species? Would it lack resistance to the pathogen?

We agree with reviewer 3 that given the requirement of cysl-1 in both *C. elegans* and C. kamaaina for the intergenerational adaptation to P. vranovensis, it would be interesting to see if increased cysl-1 expression might explain the increased naïve resistance of C. briggsae. We have now created a cysl-1 KO in *C. briggsae*. We found that cysl-1 *C. briggsae* mutants remain resistant to *P. vranovensis* even in the absence of cysl-1.

It is likely that other genes also act as modifiers of the response to *P. vranovensis* and cysl-1 activity alone is not the sole determinant of resistance in *C. briggsae*. Consistent with this hypothesis we have recently performed screens on *C. elegans* cysl-1 and cysl-2 mutants and identified mutations that render *C. elegans* resistant to *P. vranovensis* even in the absence of cysl-1 (unpublished).

As this result does not affect the conclusions of this manuscript and because the reason for *C. briggsae’s* naïve resistance to *P. vranovensis* remains unknown we have not included this result in this version of the manuscript but are happy to include it if the reviewers found it appropriate.

The authors showed that C. elegans and C. briggsae seem to share intergenerational effects of resistance to N. parisii in other species. It would be interesting to know if the mechanisms are conserved, but unfortunately this information is lacking. Similarly, it would be interesting to know if gdph-2 mediates resistance to osmolarity stress in C. kamaaina.

This new manuscript now includes RNAi data demonstrating that gpdh-2 is also required in C. kamaaina for proper adaptation to osmotic stress (Figure 1—figure supplement 1D).

The exact mechanism by which *C. elegans* adapts to N. parisii infection is not yet known so we were unable to directly test if the mechanism is conserved in C. briggsae. We have, however, previously found that loss of parental lin-35 can promote offspring resistance to N. parisii (Willis et al., 2021). We therefore created a lin-35 deletion mutant in C. briggsae to test if this same effect occurred in C. briggsae but this mutant was incredibly sick and was lost after only a few generations when no more fertile offspring were produced. We have therefore been unable to test this role of lin-35 in C. briggsae.

The supplementary tables would be more useful if the three-letter name would be included as well.

We agree and have updated the supplementary tables to include this.

Indicate in Figure 3C that these are RNAi experiments.

Done.

Also for Figure 3C, spell out "EV" in the legend of the figure. I suppose it means 'empty vector'?

EV does mean Empty Vector and we have updated the figure to now say “Empty Vector”.

The font size in the figures is pretty small and therefore difficult to read.

We have increased the font size in the figures.